# Analytic solution and stationary phase approximation for the Bayesian lasso and elastic net

**Tom Michoel**

The Roslin Institute, The University of Edinburgh, UK

Computational Biology Unit, Department of Informatics, University of Bergen, Norway

`tom.michoel@uib.no`

## Abstract

The lasso and elastic net linear regression models impose a double-exponential prior distribution on the model parameters to achieve regression shrinkage and variable selection, allowing the inference of robust models from large data sets. However, there has been limited success in deriving estimates for the full posterior distribution of regression coefficients in these models, due to a need to evaluate analytically intractable partition function integrals. Here, the Fourier transform is used to express these integrals as complex-valued oscillatory integrals over "regression frequencies". This results in an analytic expansion and stationary phase approximation for the partition functions of the Bayesian lasso and elastic net, where the non-differentiability of the double-exponential prior has so far eluded such an approach. Use of this approximation leads to highly accurate numerical estimates for the expectation values and marginal posterior distributions of the regression coefficients, and allows for Bayesian inference of much higher dimensional models than previously possible.

## 1  Introduction

Statistical modelling of high-dimensional data sets where the number of variables exceeds the number of experimental samples may result in over-fitted models that do not generalize well to unseen data. Prediction accuracy in these situations can often be improved by shrinking regression coefficients towards zero [1]. Bayesian methods achieve this by imposing a prior distribution on the regression coefficients whose mass is concentrated around zero. For linear regression, the most popular methods are ridge regression [2], which has a normally distributed prior; lasso regression [3], which has a double-exponential or Laplace distribution prior; and elastic net regression [4], whose prior interpolates between the lasso and ridge priors. The lasso and elastic net are of particular interest, because in their maximum-likelihood solutions, a subset of regression coefficients are exactly zero. However, maximum-likelihood solutions only provide a point estimate for the regression coefficients. A fully Bayesian treatment that takes into account uncertainty due to data noise and limited sample size, and provides posterior distributions and confidence intervals, is therefore of great interest.

Unsurprisingly, Bayesian inference for the lasso and elastic net involves analytically intractable partition function integrals and requires the use of numerical Gibbs sampling techniques [5–8]. However, Gibbs sampling is computationally expensive and, particularly in high-dimensional settings, convergence may be slow and difficult to assess or remedy [9–12]. An alternative to Gibbs sampling for Bayesian inference is to use asymptotic approximations to the intractable integrals based on Laplace's method [13, 14]. However, the Laplace approximation requires twice differentiable log-likelihood functions, and cannot be applied to the lasso and elastic net models as they contain a non-differentiable term proportional to the sum of absolute values (i.e. $\ell_1$-norm) of the regression coefficients.

Alternatives to the Laplace approximation have been considered for statistical models where the Fisher information matrix is singular, and no asymptotic approximation using normal distributions is feasible [15, 16]. However, in $\ell_1$-penalized models, the singularity originates from the prior distributions on the model parameters, and the Fisher information matrix remains positive definite. Here we show that in such models, approximate Bayesian inference is in fact possible using a Laplace-like approximation, more precisely the stationary phase or saddle point approximation for complex-valued oscillatory integrals [17]. This is achieved by rewriting the partition function integrals in terms of "frequencies" instead of regression coefficients, through the use of the Fourier transform. The appearance of the Fourier transform in this context should not come as a big surprise. The stationary phase approximation can be used to obtain or invert characteristic functions, which are of course Fourier transforms [18]. More to the point of this paper, there is an intimate connection between the Fourier transform of the exponential of a convex function and the Legendre-Fenchel transform of that convex function, which plays a fundamental role in physics by linking microscopic statistical mechanics to macroscopic thermodynamics and quantum to classical mechanics [19]. In particular, convex duality [20, 21], which maps the solution of a convex optimization problem to that of its dual, is essentially equivalent to writing the partition function of a Gibbs probability distribution in coordinate or frequency space (Appendix A).

Convex duality principles have been essential to characterize analytical properties of the maximum-likelihood solutions of the lasso and elastic net regression models [22–27]. This paper shows that equally powerful duality principles exist to study Bayesian inference problems.

## 2 Analytic results

We consider the usual setup for linear regression where there are $n$ observations of $p$ predictor variables and one response variable, and the effects of the predictors on the response are to be determined by minimizing the least squares cost function $\|y - Ax\|^2$ subject to additional constraints, where $y \in \mathbb{R}^n$ are the response data, $A \in \mathbb{R}^{n \times p}$ are the predictor data, $x \in \mathbb{R}^p$ are the regression coefficients which need to be estimated and $\|v\| = (\sum_{i=1}^n |v_i|^2)^{1/2}$ is the $\ell^2$-norm. Without loss of generality, it is assumed that the response and predictors are centred and standardized,

$$\sum_{i=1}^n y_i = \sum_{i=1}^n A_{ij} = 0 \quad \text{and} \quad \sum_{i=1}^n y_i^2 = \sum_{i=1}^n A_{ij}^2 = n \quad \text{for } j \in \{1, 2, \ldots, p\}. \tag{1}$$

In a Bayesian setting, a hierarchical model is assumed where each sample $y_i$ is drawn independently from a normal distribution with mean $A_{i\bullet}x$ and variance $\sigma^2$, where $A_{i\bullet}$ denotes the $i^{\text{th}}$ row of $A$, or more succinctly,

$$p(y \mid A, x) = \mathcal{N}(Ax, \sigma^2 \mathbb{1}), \tag{2}$$

where $\mathcal{N}$ denotes a multivariate normal distribution, and the regression coefficients $x$ are assumed to have a prior distribution

$$p(x) \propto \exp\left[-\frac{n}{\sigma^2}\left(\lambda\|x\|^2 + 2\mu\|x\|_1\right)\right], \tag{3}$$

where $\|x\|_1 = \sum_{j=1}^p |x_j|$ is the $\ell_1$-norm, and the prior distribution is defined upto a normalization constant. The apparent dependence of the prior distribution on the data via the dimension paramater $n$ only serves to simplify notation, allowing the posterior distribution of the regression coefficients to be written, using Bayes' theorem, as

$$p(x \mid y, A) \propto p(y \mid x, A)p(x) \propto e^{-\frac{n}{\sigma^2}\mathcal{L}(x|y,A)}, \tag{4}$$

where

$$\mathcal{L}(x \mid y, A) = \frac{1}{2n}\|y - Ax\|^2 + \lambda\|x\|^2 + 2\mu\|x\|_1 \tag{5}$$

$$= x^T\left(\frac{A^T A}{2n} + \lambda\mathbb{1}\right)x - 2\left(\frac{A^T y}{2n}\right)^T x + 2\mu\|x\|_1 + \frac{1}{2n}\|y\|^2 \tag{6}$$

is minus the posterior log-likelihood function. The maximum-likelihood solutions of the lasso ($\lambda = 0$) and elastic net ($\lambda > 0$) models are obtained by minimizing $\mathcal{L}$, where the relative scaling of the penalty

parameters to the sample size $n$ corresponds to the notational conventions of [28][1]. In the current setup, it is assumed that the parameters $\lambda \geq 0$, $\mu > 0$ and $\sigma^2 > 0$ are given a priori.

To facilitate notation, a slightly more general class of cost functions is defined as

$$H(x \mid C, w, \mu) = x^T C x - 2w^T x + 2\mu \|x\|_1, \tag{7}$$

where $C \in \mathbb{R}^{p \times p}$ is a positive-definite matrix, $w \in \mathbb{R}^p$ is an arbitrary vector and $\mu > 0$. After discarding a constant term, $\mathcal{L}(x \mid y, A)$ is of this form, as is the so-called "non-naive" elastic net, where $C = (\frac{1}{2n} A^T A + \lambda \mathbb{1})/(\lambda + 1)$ [4]. More importantly perhaps, eq. (7) also covers linear mixed models, where samples need not be independent [29]. In this case, eq. (2) is replaced by $p(y \mid A, x) = \mathcal{N}(Ax, \sigma^2 K)$, for some covariance matrix $K \in \mathbb{R}^{n \times n}$, resulting in a posterior minus log-likelihood function with $C = \frac{1}{2n} A^T K^{-1} A + \lambda \mathbb{1}$ and $w = \frac{1}{2n} A^T K^{-1} y$. The requirement that $C$ is positive definite, and hence invertible, implies that $H$ is strictly convex and hence has a unique minimizer. For the lasso ($\lambda = 0$) this only holds without further assumptions if $n \geq p$ [26]; for the elastic net ($\lambda > 0$) there is no such constraint.

The Gibbs distribution on $\mathbb{R}^p$ for the cost function $H(x \mid C, w, \mu)$ with inverse temperature $\tau$ is defined as

$$p(x \mid C, w, \mu) = \frac{e^{-\tau H(x \mid C, w, \mu)}}{Z(C, w, \mu)}.$$

For ease of notation we will henceforth drop explicit reference to $C$, $w$ and $\mu$. The normalization constant $Z = \int_{\mathbb{R}^p} e^{-\tau H(x)} dx$ is called the partition function. There is no known analytic solution for the partition function integral. However, in the posterior distribution (4), the inverse temperature $\tau = \frac{n}{\sigma^2}$ is large, firstly because we are interested in high-dimensional problems where $n$ is large (even if it may be small compared to $p$), and secondly because we assume a priori that (some of) the predictors are informative for the response variable and that therefore $\sigma^2$, the amount of variance of $y$ unexplained by the predictors, must be small. It therefore makes sense to seek an analytic approximation to the partition function for large values of $\tau$. However, the usual approach to approximate $e^{-\tau H(x)}$ by a Gaussian in the vicinity of the minimizer of $H$ and apply a Laplace approximation [17] is not feasible, because $H$ is not twice differentiable. Instead we observe that $e^{-\tau H(x)} = e^{-2\tau f(x)} e^{-2\tau g(x)}$ where

$$f(x) = \frac{1}{2} x^T C x - w^T x \tag{8}$$

$$g(x) = \mu \sum_{j=1}^{p} |x_j|. \tag{9}$$

Using Parseval's identity for Fourier transforms (Appendix A.1), it follows that (Appendix A.3)

$$Z = \int_{\mathbb{R}^p} e^{-2\tau f(x)} e^{-2\tau g(x)} dx = \frac{1}{(\pi \tau)^{\frac{p}{2}} \sqrt{\det(C)}} \int_{\mathbb{R}^p} e^{-\tau (k - iw)^T C^{-1} (k - iw)} \prod_{j=1}^{p} \frac{\mu}{k_j^2 + \mu^2} dk. \tag{10}$$

After a change of variables $z = -ik$, $Z$ can be written as a $p$-dimensional complex contour integral

$$Z = \frac{(-i\mu)^p}{(\pi \tau)^{\frac{p}{2}} \sqrt{\det(C)}} \int_{-i\infty}^{i\infty} \cdots \int_{-i\infty}^{i\infty} e^{\tau (z-w)^T C^{-1} (z-w)} \prod_{j=1}^{p} \frac{1}{\mu^2 - z_j^2} \, dz_1 \ldots dz_p. \tag{11}$$

Cauchy's theorem [30, 31] states that this integral remains invariant if the integration contours are deformed, as long as we remain in a domain where the integrand does not diverge (Appendix A.4). The analogue of Laplace's approximation for complex contour integrals, known as the stationary phase, steepest descent or saddle point approximation, then states that an integral of the form (11) can be approximated by a Gaussian integral along a steepest descent contour passing through the saddle point of the argument of the exponential function [17]. Here, the function $(z - w)^T C^{-1} (z - w)$ has a saddle point at $z = w$. If $|w_j| < \mu$ for all $j$, the standard stationary phase approximation can be applied directly, but this only covers the uninteresting situation where the maximum-likelihood solution $\hat{x} = \operatorname{argmin}_x H(x) = 0$ (Appendix A.5). As soon as $|w_j| > \mu$ for at least one $j$, the standard

argument breaks down, since to deform the integration contours from the imaginary axes to parallel contours passing through the saddle point $z_0 = w$, we would have to pass through a pole (divergence) of the function $\prod_j (\mu^2 - z_j^2)^{-1}$ (Figure S1). Motivated by similar, albeit one-dimensional, analyses in non-equilibrium physics [32, 33], we instead consider a temperature-dependent function

$$H_\tau^*(z) = (z - w)^T C^{-1}(z - w) - \frac{1}{\tau} \sum_{j=1}^p \ln(\mu^2 - z_j^2), \tag{12}$$

which is well-defined on the domain $\mathcal{D} = \{z \in \mathbb{C}^p \colon |\Re z_j| < \mu, \ j = 1, \ldots, p\}$, where $\Re$ denotes the real part of a complex number. This function has a unique saddle point in $\mathcal{D}$, regardless whether $|w_j| < \mu$ or not (Figure S1). Our main result is a steepest descent approximation of the partition function around this saddle point.

**Theorem 1** *Let $C \in \mathbb{R}^{p \times p}$ be a positive definite matrix, $w \in \mathbb{R}^p$ and $\mu > 0$. Then the complex function $H_\tau^*$ defined in eq. (12) has a unique saddle point $\hat{u}_\tau$ that is real, $\hat{u}_\tau \in \mathcal{D} \cap \mathbb{R}^p$, and is a solution of the set of third order equations*

$$(\mu^2 - u_j^2)[C^{-1}(w - u)]_j - \frac{u_j}{\tau} = 0 \,, \quad u \in \mathbb{R}^p, \ j \in \{1, \ldots, p\}. \tag{13}$$

*For $Q(z)$ a complex analytic function of $z \in \mathbb{C}^p$, the generalized partition function*

$$Z[Q] = \frac{1}{(\pi\tau)^{\frac{p}{2}} \sqrt{\det(C)}} \int_{\mathbb{R}^p} e^{-\tau(k - iw)^T C^{-1}(k - iw)} Q(-ik) \prod_{j=1}^p \frac{\mu}{k_j^2 + \mu^2} dk.$$

*can be analytically expressed as*

$$Z[Q] = \left(\frac{\mu}{\sqrt{\tau}}\right)^p e^{\tau(w - \hat{u}_\tau)^T C^{-1}(w - \hat{u}_\tau)} \prod_{j=1}^p \frac{1}{\sqrt{\mu^2 + \hat{u}_{\tau,j}^2}} \frac{1}{\sqrt{\det(C + D_\tau)}}$$

$$\exp\left\{\frac{1}{4\tau^2} \Delta_\tau\right\} e^{R_\tau(ik)} Q(\hat{u}_\tau + ik) \bigg|_{k=0}, \quad (14)$$

*where $D_\tau$ is a diagonal matrix with diagonal elements*

$$(D_\tau)_{jj} = \frac{\tau(\mu^2 - \hat{u}_{\tau,j}^2)^2}{\mu^2 + \hat{u}_{\tau,j}^2}, \tag{15}$$

$\Delta_\tau$ *is the differential operator*

$$\Delta_\tau = \sum_{i,j=1}^p \left[\tau D_\tau (C + D_\tau)^{-1} C\right]_{ij} \frac{\partial^2}{\partial k_i \partial k_j} \tag{16}$$

*and*

$$R_\tau(z) = \sum_{j=1}^p \sum_{m \geq 3} \frac{1}{m} \left[\frac{1}{(\mu - \hat{u}_{\tau,j})^m} + \frac{(-1)^m}{(\mu + \hat{u}_{\tau,j})^m}\right] z_j^m. \tag{17}$$

*This results in an analytic approximation*

$$Z[Q] \sim \left(\frac{\mu}{\sqrt{\tau}}\right)^p e^{\tau(w - \hat{u}_\tau)^T C^{-1}(w - \hat{u}_\tau)} \prod_{j=1}^p \frac{1}{\sqrt{\mu^2 + \hat{u}_{\tau,j}^2}} \frac{Q(\hat{u}_\tau)}{\sqrt{\det(C + D_\tau)}}. \tag{18}$$

The analytic expression in eq. (14) follows by changing the integration contours to pass through the saddle point $\hat{u}_\tau$, and using a Taylor expansion of $H_\tau^*(z)$ around the saddle point along the steepest descent contour. However, because $\Delta_\tau$ and $R_\tau$ depend on $\tau$, it is not a priori evident that (18) holds. A detailed proof is given in Appendix B. The analytic approximation in eq. (18) can be simplified further by expanding $\hat{u}_\tau$ around its leading term, resulting in an expression that recognizably converges to the sparse maximum-likelihood solution (Appendix C). While eq. (18) is computationally more expensive to calculate than the corresponding expression in terms of the maximum-likelihood solution, it was found to be numerically more accurate (Section 3).

Various quantities derived from the posterior distribution can be expressed in terms of generalized partition functions. The most important of these are the expectation values of the regression coefficients, which, using elementary properties of the Fourier transform (Appendix A.6), can be expressed as

$$\mathbb{E}(x) = \frac{1}{Z} \int_{\mathbb{R}^p} x \, e^{-\tau H(x)} = \frac{Z\big[C^{-1}(w-z)\big]}{Z} \sim C^{-1}(w - \hat{u}_\tau).$$

The leading term,

$$\hat{x}_\tau \equiv C^{-1}(w - \hat{u}_\tau), \tag{19}$$

can be interpreted as an estimator for the regression coefficients in its own right, which interpolates smoothly (as a function of $\tau$) between the ridge regression estimator $\hat{x}_{\text{ridge}} = C^{-1}w$ at $\tau = 0$ and the maximum-likelihood elastic net estimator $\hat{x} = C^{-1}(w - \hat{u})$ at $\tau = \infty$, where $\hat{u} = \lim_{\tau \to \infty} \hat{u}_\tau$ satisfies a box-constrained optimization problem (Appendix C).

The marginal posterior distribution for a subset $I \subset \{1, \ldots, p\}$ of regression coefficients is defined as

$$p(x_I) = \frac{1}{Z(C,w,\mu)} \int_{\mathbb{R}^{|I^c|}} e^{-\tau H(x|C,w,\mu)} dx_{I^c}$$

where $I^c = \{1, \ldots, p\} \setminus I$ is the complement of $I$, $|I|$ denotes the size of a set $I$, and we have reintroduced temporarily the dependency on $C$, $w$ and $\mu$ as in eq. (7). A simple calculation shows that the remaining integral is again a partition function of the same form, more precisely:

$$p(x_I) = e^{-\tau(x_I^T C_I x_I - 2w_I^T x_I + 2\mu\|x_I\|_1)} \frac{Z(C_{I^c}, w_{I^c} - x_I^T C_{I,I^c}, \mu)}{Z(C,w,\mu)}, \tag{20}$$

where the subscripts $I$ and $I^c$ indicate sub-vectors and sub-matrices on their respective coordinate sets. Hence the analytic approximation in eq. (14) can be used to approximate numerically each term in the partition function ratio and obtain an approximation to the marginal posterior distributions.

The posterior predictive distribution [1] for a new sample $a \in \mathbb{R}^p$ of predictor data can also be written as a ratio of partition functions:

$$p(y) = \int_{\mathbb{R}^p} p(y \mid a, x) p(x \mid C, w, \mu) \, dx = \left(\frac{\tau}{2\pi n}\right)^{\frac{1}{2}} e^{-\frac{\tau}{2n}y^2} \frac{Z\big(C + \frac{1}{2n}aa^T, w + \frac{y}{2n}a, \mu\big)}{Z(C,w,\mu)},$$

where $C \in \mathbb{R}^{p \times p}$ and $w \in \mathbb{R}^p$ are obtained from the training data as before, $n$ is the number of training samples, and $y \in \mathbb{R}$ is the unknown response to $a$ with distribution $p(y)$. Note that

$$\mathbb{E}(y) = \int_{\mathbb{R}} y p(y) dy = \int_{\mathbb{R}^p} \left[\int_{\mathbb{R}} p(y \mid a, x) dy\right] p(x \mid C, w, \mu) \, dx$$

$$= \int_{\mathbb{R}^p} a^T x p(x \mid C, w, \mu) \, dx = a^T \mathbb{E}(x) \sim a^T \hat{x}_\tau. \tag{21}$$

## 3 Numerical experiments

To test the accuracy of the stationary phase approximation, we implemented algorithms to solve the saddle point equations and compute the partition function and marginal posterior distribution, as well as an existing Gibbs sampler algorithm [8] in Matlab (see Appendix E for algorithm details, source code available from https://github.com/tmichoel/bayonet/). Results were first evaluated for independent predictors (or equivalently, one predictor) and two commonly used data sets: the "diabetes data" ($n = 442$, $p = 10$) [34] and the "leukemia data" ($n = 72$, $p = 3571$) [4] (see Appendix F for further experimental details and data sources).

First we tested the rate of convergence in the asymptotic relation (see Appendix C)

$$\lim_{\tau \to \infty} -\frac{1}{\tau} \log Z = H_{\min} = \min_{x \in \mathbb{R}^p} H(x).$$

For independent predictors ($p = 1$), the partition function can be calculated analytically using the error function (Appendix D), and rapid convergence to $H_{\min}$ is observed (Figure 1a). After scaling by the number of predictors $p$, a similar rate of convergence is observed for the stationary

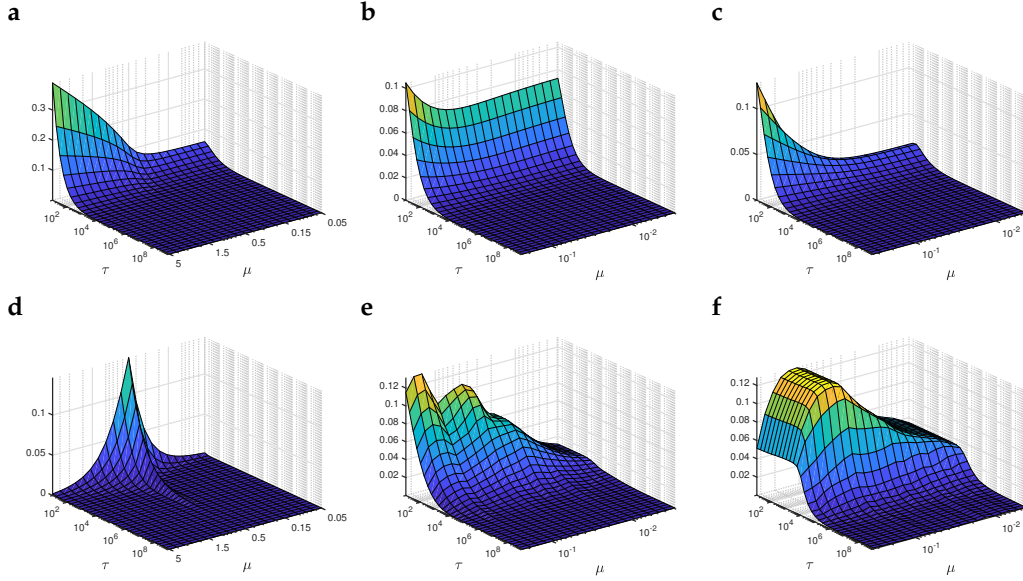

Figure 1: Convergence to the minimum-energy solution. Top row: $(-\frac{1}{\tau}\log Z - H_{\min})/p$ vs. $\tau$ and $\mu$ for the exact partition function for independent predictors ($p = 1$) (**a**), and for the stationary phase approximation for the diabetes (**b**) and leukemia (**c**) data. Bottom row: $\|\hat{x}_\tau - \hat{x}\|_\infty$ for the exact expectation value for independent predictors (**d**), and using the stationary phase approximation for the diabetes (**e**) and leukemia (**f**) data. Parameter values were $C = 1.0$, $w = 0.5$, and $\mu$ ranging from 0.05 to 5 in geometric steps (**a**), and $\lambda = 0.1$ and $\mu$ ranging from $0.01\mu_{\max}$ upto, but not including, $\mu_{\max} = \max_j |w_j|$ in geometric steps (**b**,**c**).

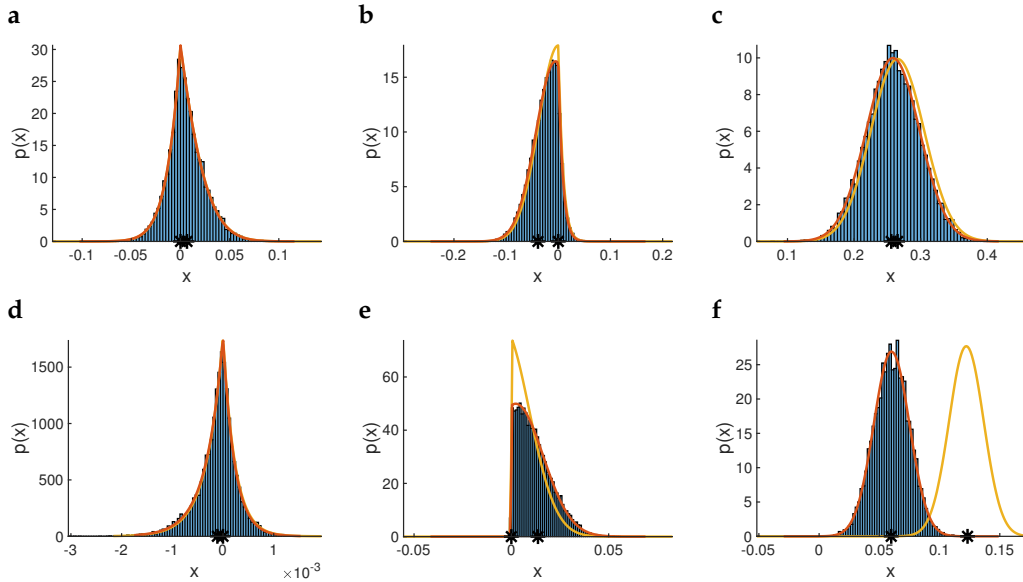

Figure 2: Marginal posterior distributions for the diabetes data ($\lambda = 0.1$, $\mu = 0.0397$, $\tau = 682.3$) (**a–c**) and leukemia data ($\lambda = 0.1$, $\mu = 0.1835$, $\tau = 9943.9$) (**d–f**;). In blue, Gibbs sampling histogram ($10^4$ samples). In red, stationary phase approximation for the marginal posterior distribution of selected predictors. In yellow, maximum-likelihood-based approximation for the same distributions. The distributions for a zero, transition and non-zero maximum-likelihood predictor are shown (from left to right). The $*$ on the $x$-axes indicate the location of the maximum-likelihood and posterior expectation value.

phase approximation to the partition function for both the diabetes and leukemia data (Figure 1b,c). However, convergence of the posterior expectation values $\hat{x}_\tau$ to the maximum-likelihood coefficients $\hat{x}$, as measured by the $\ell_\infty$-norm difference $\|\hat{x}_\tau - \hat{x}\|_\infty = \max_j |\hat{x}_{\tau,j} - \hat{x}_j|$ is noticeably slower, particularly in the $p \gg n$ setting of the leukemia data (Figure 1d–f).

Next, the accuracy of the stationary phase approximation at finite $\tau$ was determined by comparing the marginal distributions for single predictors [i.e. where $I$ is a singleton in eq. (20)] to results obtained from Gibbs sampling. For simplicity, representative results are shown for specific hyper-parameter values (Appendix F.2). Application of the stationary phase approximation resulted in marginal posterior distributions which were indistinguishable from those obtained by Gibbs sampling (Figure 2). In view of the convergence of the log-partition function to the minimum-energy value (Figure 1), an approximation to eq. (20) of the form

$$p(x_I) \approx e^{-\tau(x_I^T C_I x_I - 2w_I^T x_I + 2\mu\|x_I\|_1)} e^{-\tau[H_{\min}(C_{I^c}, w_{I^c} - x_I^T C_{I,I^c}, \mu) - H_{\min}(C, w, \mu)]} \quad (22)$$

was also tested. However, while eq. (22) is indistinguishable from eq. (20) for predictors with zero effect size in the maximum-likelihood solution, it resulted in distributions that were squeezed towards zero for transition predictors, and often wildly inaccurate for non-zero predictors (Figure 2). This is because eq. (22) is maximized at $x_I = \hat{x}_I$, the maximum-likelihood value, whereas for non-zero coordinates, eq. (20) is (approximately) symmetric around its expectation value $\mathbb{E}(x_I) = \hat{x}_{\tau,I}$. Hence, accurate estimations of the marginal posterior distributions requires using the full stationary phase approximations [eq. (18)] to the partition functions in eq. (20).

The stationary phase approximation can be particularly advantageous in prediction problems, where the response value $\hat{y} \in \mathbb{R}$ for a newly measured predictor sample $a \in \mathbb{R}^p$ is obtained using regression coefficients learned from training data $(y_t, A_t)$. In Bayesian inference, $\hat{y}$ is set to the expectation value of the posterior predictive distribution, $\hat{y} = \mathbb{E}(y) = a^T \hat{x}_\tau$ [eq. (21)]. Computation of the posterior expectation values $\hat{x}_\tau$ [eq. (19)] using the stationary phase approximation requires solving only one set of saddle point equations, and hence can be performed efficiently across a range of hyper-parameter values, in contrast to Gibbs sampling, where the full posterior needs to be sampled even if only expectation values are needed.

To illustrate how this benefits large-scale applications of the Bayesian elastic net, its prediction performance was compared to state-of-the-art Gibbs sampling implementations of Bayesian horseshoe and Bayesian lasso regression [35], as well as to maximum-likelihood elastic net and ridge regression, using gene expression and drug sensitivity data for 17 anticancer drugs in 474 human cancer cell lines from the Cancer Cell Line Encyclopedia [36]. Ten-fold cross-validation was used, using $p = 1000$ pre-selected genes and 427 samples for training regression coefficients and 47 for validating predictions in each fold. To obtain unbiased predictions at a single choice for the hyper-parameters, $\mu$ and $\tau$ were optimized over a $10 \times 13$ grid using an additional internal 10-fold cross-validation on the training data only (385 samples for training, 42 for testing); BayReg's lasso and horseshoe methods sample hyper-parameter values from their posteriors and do not require an additional cross-validation loop (see Appendix F.3 for complete experimental details and data sources). Despite evaluating a much greater number of models (in each cross-validation fold, $10\times$ cross-validation over 130 hyper-parameter combinations vs. 1 model per fold), the overall computation time was still much lower than BayReg's Gibbs sampling approach (on average 30 sec. per fold, i.e. 0.023 sec. per model, vs. 44 sec. per fold for BayReg). In terms of predictive performance, Bayesian methods tended to perform better than maximum-likelihood methods, in particular for the most 'predictable' responses, with little variation between the three Bayesian methods (Figure 3a).

While the difference in optimal performance between Bayesian and maximum-likelihood elastic net was not always large, Bayesian elastic net tended to be optimized at larger values of $\mu$ (i.e. at sparser maximum-likelihood solutions), and at these values the performance improvement over maximum-likelihood elastic net was more pronounced (Figure 3b). As expected, $\tau$ acts as a tuning parameter that allows to smoothly vary from the maximum-likelihood solution at large $\tau$ (here, $\tau \sim 10^6$) to the solution with best cross-validation performance (here, $\tau \sim 10^3 - 10^4$) (Figure 3c). The improved performance at sparsity-inducing values of $\mu$ suggests that the Bayesian elastic net is uniquely able to identify the dominant predictors for a given response (the non-zero maximum-likelihood coefficients), while still accounting for the cumulative contribution of predictors with small effects. Comparison with the unpenalized ($\mu = 0$) ridge regression coefficients shows that the Bayesian expectation values are strongly shrunk towards zero, except for the non-zero maximum-likelihood coefficients, which remain relatively unchanged (Figure 3d), resulting in a double-exponential distribution for

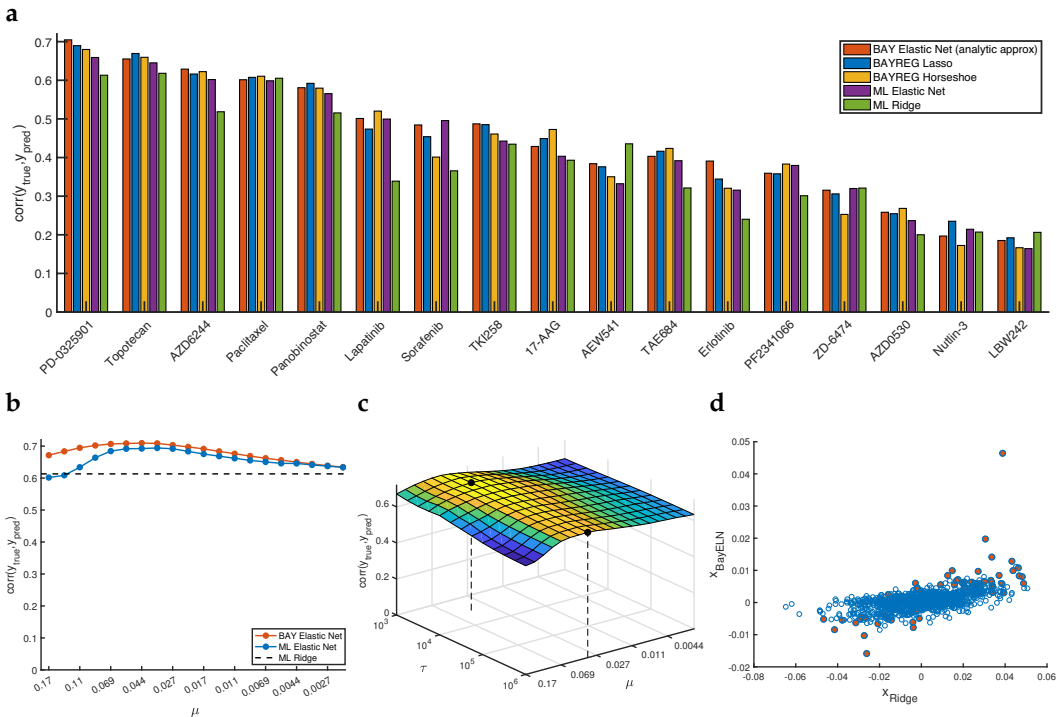

Figure 3: Predictive accuracy on the Cancer Cell Line Encyclopedia. **a.** Median correlation coefficient between predicted and true drug sensitivities over 10-fold cross-validation, using Bayesian posterior expectation values from the analytic approximation for elastic net (red) and from BayReg's lasso (blue) and horseshoe (yellow) implementations, and maximum-likelihood elastic net (purple) and ridge regression (green) values for the regression coefficients. See main text for details on hyper-parameter optimization. **b.** Median 10-fold cross-validation value for the correlation coefficient between predicted and true sensitivities for the compound PD-0325901 vs. $\mu$, for the Bayesian elastic net at optimal $\tau$ (red), maximum-likelihood elastic net (blue) and ridge regression (dashed). **c.** Median 10-fold cross-validation value for the correlation coefficient between predicted and true sensitivities for PD-0325901 for the Bayesian elastic net vs. $\tau$ and $\mu$; the black dots show the overall maximum and the ML maximum. **d.** Scatter plot of expected regression coefficients in the Bayesian elastic net for PD-0325901 at $\mu = 0.055$ and optimal $\tau = 3.16 \cdot 10^3$ vs. ridge regression coefficient estimates; coefficients with non-zero maximum-likelihood elastic net value at the same $\mu$ are indicated in red. See Supp. Figures S2 and S3 for the other 16 compounds.

the regression coefficients. This contrasts with ridge regression, where regression coefficients are normally distributed leading to over-estimation of small effects, and maximum-likelihood elastic net, where small effects become identically zero and don't contribute to the predicted value at all.

## 4    Conclusions

The application of Bayesian methods to infer expected effect sizes and marginal posterior distributions in $\ell_1$-penalized models has so far required the use of computationally expensive Gibbs sampling methods. Here it was shown that highly accurate inference in these models is also possible using an analytic stationary phase approximation to the partition function integrals. This approximation exploits the fact that the Fourier transform of the non-differentiable double-exponential prior distribution is a well-behaved exponential of a log-barrier function, which is intimately related to the Legendre-Fenchel transform of the $\ell_1$-penalty term. Thus, the Fourier transform plays the same role for Bayesian inference problems as convex duality plays for maximum-likelihood approaches.

For simplicity, we have focused on the linear regression model, where the invariance of multivariate normal distributions under the Fourier transform greatly facilitates the analytic derivations. Preliminary work shows that the results can probably be extended to generalized linear models (or any

model with convex energy function) with L1 penalty, using the argument sketched in Appendix A.2. In such models, the predictor correlation matrix $C$ will need to be replaced by the Hessian matrix of the energy function evaluated at the saddle point. Numerically, this will require updates of the Hessian during the coordinate descent algorithm for solving the saddle point equations. How to balance the accuracy of the approximation and the frequency of the Hessian updates will require further in-depth investigation. In principle, the same analysis can also be performed using other non-twice-differentiable sparse penalty functions, but if their Fourier transform is not known analytically, or not twice differentiable either, the analysis and implementation will become more complicated still.

A limitation of the current approach may be that values of the hyper-parameters need to be specified in advance, whereas in complete hierarchical models, these are subject to their own prior distributions. Incorporation of such priors will require careful attention to the interchange between taking the limit of and integrating over the inverse temperature parameter. However, in many practical situations $\ell_1$ and $\ell_2$-penalty parameters are pre-determined by cross-validation. Setting the residual variance parameter to its maximum a-posteriori value then allows to evaluate the maximum-likelihood solution in the context of the posterior distribution of which it is the mode [8]. Alternatively, if the posterior expectation values of the regression coefficients are used instead of their maximum-likelihood values to predict unmeasured responses, the optimal inverse-temperature parameter can be determined by standard cross-validation on the training data, as in the drug response prediction experiments.

No attempt was made to optimize the efficiency of the coordinate descent algorithm to solve the saddle point equations. However, comparison to the Gibbs sampling algorithm shows that one cycle through all coordinates in the coordinate descent algorithm is approximately equivalent to one cycle in the Gibbs sampler, i.e. to adding one more sample. The coordinate descent algorithm typically converges in 5-10 cycles starting from the maximum-likelihood solution, and 1-2 cycles when starting from a neighbouring solution in the estimation of marginal distributions. In contrast, Gibbs sampling typically requires $10^3$-$10^5$ coordinate cycles to obtain stable distributions. Hence, if only the posterior expectation values or the posterior distributions for a limited number of coordinates are sought, the computational advantage of the stationary phase approximation is vast. On the other hand, each evaluation of the marginal distribution functions requires the solution of a separate set of saddle point equations. Hence, computing these distributions for all predictors at a very large number of points with the current algorithm could become equally expensive as Gibbs sampling.

In summary, expressing intractable partition function integrals as complex-valued oscillatory integrals through the Fourier transform is a powerful approach for performing Bayesian inference in the lasso and elastic net regression models, and $\ell_1$-penalized models more generally. Use of the stationary phase approximation to these integrals results in highly accurate estimates for the posterior expectation values and marginal distributions at a much reduced computational cost compared to Gibbs sampling.

**Acknowledgments**

This research was supported by the BBSRC (grant numbers BB/J004235/1 and BB/M020053/1).

## Footnotes

[1]To be precise, in [28] the penalty term is written as $\tilde{\lambda}(\frac{1-\alpha}{2} \|x\|_2^2 + \alpha \|x\|_1)$, wich is obtained from (5) by setting $\tilde{\lambda} = 2(\lambda + \mu)$ and $\alpha = \frac{\mu}{\lambda + \mu}$.

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
