[Supplementary Material]

# Analytic solution and stationary phase approximation for the Bayesian lasso and elastic net

**Tom Michoel**
The Roslin Institute, The University of Edinburgh, UK
Computational Biology Unit, Department of Informatics, University of Bergen, Norway
`tom.michoel@uib.no`

# Appendices

## A  Basic results in Fourier space

### A.1  Fourier transform conventions

Fourier transforms are defined with different scaling conventions in different branches of science. Here, the symmetric version of the Fourier transform written in terms of angular frequencies is used: for $f$ a function on $\mathbb{R}^p$, we define

$$\mathcal{F}[f](k) = \hat{f}(k) = \frac{1}{(2\pi)^{\frac{p}{2}}} \int_{\mathbb{R}^p} f(x) e^{-ik^T x} dx$$

and

$$f(x) = \mathcal{F}^{-1}\big[\mathcal{F}[f]\big](x) = \frac{1}{(2\pi)^{\frac{p}{2}}} \int_{\mathbb{R}^p} \hat{f}(k) e^{ik^T x} dk.$$

Parseval's identity states that for two functions $f$ and $g$,

$$\int_{\mathbb{R}^p} \overline{f(x)} g(x) dx = \int_{\mathbb{R}^p} \overline{\hat{f}(k)} \hat{g}(k) dk,$$

where $\bar{\cdot}$ denotes complex conjugation. For more details, see [1, Chapter 11].

### A.2  Relation between convex duality and the Fourier transform

The motivation for using the Fourier transform to study Bayesian inference problems stems from the correspondence between the Fourier and Legendre-Fenchel transforms of convex functions. This correspondence is an example of so-called idempotent mathematics, and a survey of its history and applications can be found in [2], while a formal treatment along the lines below can be found in [3], and a summary of analogous properties between the Legendre-Fenchel and Fourier transforms can be found in [4]. The basic argument is presented here, without any attempt at being complete or rigorous.

Let $h$ be a convex function on $\mathbb{R}^p$ and assume it is sufficiently smooth for the statements below to hold without needing too much attention to detail. The Gibbs probability distribution for $h$ at inverse temperature $\tau$ is defined as $p(x) = \frac{1}{Z} e^{-\tau h(x)}$, with $Z = \int_{\mathbb{R}^p} e^{-\tau h(x)} dx$ the partition function. Define for $z \in \mathbb{C}^p$

$$h_\tau^*(z) = \frac{1}{\tau} \ln \int_{\mathbb{R}^p} e^{-\tau[h(x) - z^T x]} dx.$$

By the Laplace approximation, it follows that for $\tau$ large and $u \in \mathbb{R}^p$, to leading order in $\tau$,

$$h_\tau^*(u) \approx h^*(u) = \max_{x \in \mathbb{R}^p}[u^T x - h(x)], \tag{1}$$

the Legendre-Fenchel transform of $h$. The Fourier transform of $e^{-\tau h}$ is

$$\mathcal{F}\big[e^{-\tau h}\big](\tau k) = \frac{1}{(2\pi)^{\frac{p}{2}}} \int_{\mathbb{R}^p} e^{-\tau h(x)} e^{-i\tau k^T x} dx = \frac{e^{\tau h_\tau^*(-ik)}}{(2\pi)^{\frac{p}{2}}}. \tag{2}$$

Now assume that $h = f + g$ can be written as the sum of two convex functions $f$ and $g$. It is instructive to think of $h(x)$ as minus a posterior log-likelihood function of regression coefficients $x$, with a natural decomposition in a part $f(x)$ coming from the data likelihood and a part $g(x)$ representing the prior distribution on $x$. We again assume that $f$ and $g$ are smooth.

The Parseval identity for Fourier transforms yields

$$\int_{\mathbb{R}^p} e^{-\tau[f(x)+g(x)]} dx = \int_{\mathbb{R}^p} \overline{\mathcal{F}\big[e^{-\tau f}\big](k)} \mathcal{F}\big[e^{-\tau g}\big](k) dk = \left(\frac{\tau}{2\pi}\right)^p \int_{\mathbb{R}^p} e^{\tau[f_\tau^*(ik)+g_\tau^*(-ik)]} dk,$$

where a change of variables $k \to \tau k$ was made. When $\tau$ is large, the Laplace approximation of the l.h.s. states that, to leading order in $\tau$

$$\frac{1}{\tau} \ln \int_{\mathbb{R}^p} e^{-\tau[f(x)+g(x)]} dx \approx - \min_{x \in \mathbb{R}^p}\big[f(x) + g(x)\big] = \max_{x \in \mathbb{R}^p}\big[-f(x) - g(x)\big]. \tag{3}$$

The integral on the r.h.s. can be written as a complex contour integral

$$\int_{\mathbb{R}^p} e^{\tau[f_\tau^*(ik)+g_\tau^*(-ik)]} dk = \frac{1}{i^p} \int_{i\mathbb{R}^p} e^{\tau[f_\tau^*(z)+g_\tau^*(-z)]} dz,$$

where $i\mathbb{R}^p$ denotes a $p$-dimensional contour consisting of vertical contours running along the imaginary axis in each dimension. The steepest descent or saddle point approximation [5] requires that we deform the contour to run through the saddle point, i.e. a zero of the gradient function $\nabla[f_\tau^*(z) + g_\tau^*(-z)]$. Under fairly general conditions (see for instance [6]), $f_\tau^*(z) + g_\tau^*(-z)$ will attain its maximum modulus at a real vector, and hence the new integration contour will take the form $z = \hat{u}_\tau + ik$ where $\hat{u}_\tau = \operatorname{argmin}_{u \in \mathbb{R}^p}[f_\tau^*(u) + g_\tau^*(-u)]$ and $k \in \mathbb{R}^p$. Note that in the limit $\tau \to \infty$, $\hat{u}_\tau \to \hat{u} = \operatorname{argmin}_{u \in \mathbb{R}^p}[f^*(u) + g^*(-u)]$. The stationary phase approximation yields, again to leading order in $\tau$

$$\frac{1}{\tau} \ln \int_{\mathbb{R}^p} e^{\tau[f_\tau^*(ik)+g_\tau^*(-ik)]} dk = \frac{1}{\tau} \ln \int_{\mathbb{R}^p} e^{\tau[f_\tau^*(\hat{u}_\tau+ik)+g_\tau^*(-\hat{u}_\tau-ik)]} dk$$

$$\approx \min_{u \in \mathbb{R}^p}\big[f_\tau^*(u) + g_\tau^*(-u)\big] \approx \min_{u \in \mathbb{R}^p}\big[f^*(u) + g^*(-u)\big] \tag{4}$$

Combining eqs. (3) and (4), we recover Fenchel's well-known duality theorem

$$\max_{x \in \mathbb{R}^p}\big[-f(x) - g(x)\big] = \min_{u \in \mathbb{R}^p}\big[f^*(u) + g^*(-u)\big].$$

In summary, there is an equivalence between convex duality for log-likelihood functions and switching from coordinate to frequency space using the Fourier transform for Gibbs probability distributions, which becomes an exact mapping in the limit of large inverse temperature. As shown in this paper, this remains true even when $f$ or $g$ are not necessarily smooth (e.g. if $g(x) = \|x\|_1$ is the $\ell_1$-norm).

### A.3   The Fourier transform of the multivariate normal and Laplace distributions

To derive eq. (10), observe that $f(x)$ is a Gaussian and its Fourier transform is again a Gaussian:

$$\overline{\mathcal{F}(e^{-2\tau f})} = \frac{1}{(2\pi)^{\frac{p}{2}}} \int_{\mathbb{R}^p} e^{-2\tau f(x)} e^{ik^T x} dx = \frac{1}{\sqrt{(2\tau)^p \det(C)}} \exp\left\{-\frac{1}{4\tau}(k - 2i\tau w)^T C^{-1}(k - 2i\tau w)\right\}. \tag{5}$$

To calculate the Fourier transform of $e^{-\tau g}$, note that in one dimension

$$\int_{\mathbb{R}} e^{-\gamma|x|} e^{-ikx} dx = \frac{2\gamma}{k^2 + \gamma^2},$$

and hence

$$\mathcal{F}(e^{-2\tau g})(k) = \frac{1}{(2\pi)^{\frac{p}{2}}} \prod_{j=1}^p \frac{4\mu\tau}{k_j^2 + 4\tau^2\mu^2}.$$

After making the change of variables $k_j' = \frac{1}{2\tau} k_j$, eq. (10) is obtained.

## A.4 Cauchy's theorem in coordinate space

Cauchy's theorem [7, 8] states that we can freely deform the integration contours in the integral in eq. (11) as long as we remain within a holomorphic domain of the integrand, or simply put, a domain where the integrand does not diverge. Consider as a simple example the deformation of the integration contours from $z_j \in i\mathbb{R}$ in eq. (11) to $z_j \in w'_j + i\mathbb{R}$, where $|w'_j| < \mu$ for all $j$. We obtain

$$
Z = \frac{(-i\mu)^p}{(\pi\tau)^{\frac{p}{2}}\sqrt{\det(C)}} \int_{w'_1-i\infty}^{w'_1+i\infty} \cdots \int_{w'_p-i\infty}^{w'_p+i\infty} e^{\tau(z-w)^T C^{-1}(z-w)} \prod_{j=1}^{p} \frac{1}{\mu^2 - z_j^2} \, dz_1 \ldots dz_p
$$

$$
= \frac{\mu^p}{(\pi\tau)^{\frac{p}{2}}\sqrt{\det(C)}} \int_{\mathbb{R}^p} e^{-\tau(w'-w+ik)^T C^{-1}(w'-w+ik)} \prod_{j=1}^{p} \frac{1}{\mu^2 - (w'_j + ik_j)^2} \, dk,
$$

where we parameterized $z_j = w'_j + ik_j$. Using the inverse Fourier transform, and reversing the results from Section 2 and Appendix A.3, we can write this expression as

$$
Z = \int_{\mathbb{R}^p} e^{-2\tau \tilde{f}(x)} e^{-2\tau \tilde{g}(x)},
$$

where

$$
\tilde{f}(x) = \frac{1}{2} x^T C x - (w - w')^T x \tag{6}
$$

$$
\tilde{g}(x) = \sum_{j=1}^{p} (\mu|x_j| - w'_j x_j). \tag{7}
$$

Comparison with eqs. (8)–(9) shows that the freedom to deform the integration contour in Fourier space corresponds to an equivalent freedom to split $e^{-\tau H(x)}$ into a product of two functions. Clearly eq. (7) only defines an integrable function $e^{-2\tau \tilde{g}}$ if $|w'_j| < \mu$ for all $j$, which of course corresponds to the limitation imposed by Cauchy's theorem that the deformation of the integration contours cannot extend beyond the domain where the function $\prod_j (\mu^2 - z_j^2)^{-1}$ remains finite.

## A.5 Stationary phase approximation in the zero-effect case

Assume that $|w_j| < \mu$ for all $j$. It then follows immediately that the maximum-likelihood or minimum-energy solution $\hat{x} = \operatorname{argmin}_x H(x) = 0$. As above, we can deform the integration contours in (11) into steepest descent contours passing through the saddle point $z_0 = w$ of the function $h(z) = (z-w)^T C^{-1}(z-w)$ (cf. Figure S1a). We obtain

$$
Z = \frac{(-i\mu)^p}{(\pi\tau)^{\frac{p}{2}}\sqrt{\det(C)}} \int_{w_1-i\infty}^{w_1+i\infty} \cdots \int_{w_p-i\infty}^{w_p+i\infty} e^{\tau(z-w)^T C^{-1}(z-w)} \prod_{j=1}^{p} \frac{1}{\mu^2 - z_j^2} \, dz_1 \ldots dz_p
$$

$$
= \frac{\mu^p}{(\pi\tau)^{\frac{p}{2}}\sqrt{\det(C)}} \int_{\mathbb{R}^p} e^{-\tau k^T C^{-1} k} \prod_{j=1}^{p} \frac{1}{\mu^2 - (w_j + ik_j)^2} \, dk, \tag{8}
$$

where we parameterized $z_j = w_j + ik_j$. This integral can be written as a series expansion using the following standard result, included here for completeness.

**Lemma 1.** *Let $C \in \mathbb{R}^p \times \mathbb{R}^p$ be a positive definite matrix and let $\Delta_C$ be the differential operator*

$$
\Delta_C = \sum_{i,j=1}^{p} C_{ij} \frac{\partial^2}{\partial k_i \partial k_j}.
$$

*Then*

$$
\frac{1}{\pi^{\frac{p}{2}}\sqrt{\det(C)}} \int_{\mathbb{R}^p} e^{-k^T C^{-1} k} \hat{f}(k) dk = \left( e^{\frac{1}{4}\Delta_C} \hat{f} \right)(0).
$$

*Proof.* First note that

$$
\Delta_C e^{-ik^T x} = -\sum_{ij} C_{ij} x_i x_j e^{-ik^T x} = -(x^T C x) e^{-ik^T x}, \tag{9}
$$

i.e. $e^{ik^Tx}$ is an 'eigenfunction' of $\Delta_C$ with eigenvalue $-(x^TCx)$, and hence

$$e^{\frac{1}{4}\Delta_C}e^{-ik^Tx} = e^{-\frac{1}{4}x^TCx}e^{-ik^Tx}.$$

Using the (inverse) Fourier transform, we can define

$$f(x) = \frac{1}{(2\pi)^{\frac{p}{2}}}\int_{\mathbb{R}^p}\hat{f}(k)e^{ik^Tx}dk,$$

and write

$$\hat{f}(k) = \frac{1}{(2\pi)^{\frac{p}{2}}}\int_{\mathbb{R}^p}f(x)e^{-ik^Tx}dx.$$

Hence

$$\left(e^{\frac{1}{4}\Delta_C}\hat{f}\right)(k) = \frac{1}{(2\pi)^{\frac{p}{2}}}\int_{\mathbb{R}^p}f(x)e^{\frac{1}{4}\Delta_C}e^{ik^Tx}dx = \frac{1}{(2\pi)^{\frac{p}{2}}}\int_{\mathbb{R}^p}f(x)e^{-\frac{1}{4}x^TCx}e^{-ik^Tx}dx.$$

Using Parseval's identity and the formula for the Fourier transform of a Gaussian [eq. (5)], we obtain

$$\left(e^{\frac{1}{4}\Delta_C}\hat{f}\right)(0) = \frac{1}{(2\pi)^{\frac{p}{2}}}\int_{\mathbb{R}^p}f(x)e^{-\frac{1}{4}x^TCx}dx = \frac{1}{\pi^{\frac{p}{2}}\sqrt{\det(C)}}\int_{\mathbb{R}^p}\hat{f}(k)e^{-k^TC^{-1}k}dk$$

$\square$

In the derivation above, we have tacitly assumed that the inverse Fourier transform $f$ of $\hat{f}$ exists. However, the result remains true even if $f$ is only a distribution, i.e. $\hat{f}$ need not be integrable. For a more detailed discussion, see [1, Chapter 11, Section 11.9].

Applying Lemma 1 to eq. (8), it follows that

$$Z = \left(\frac{\mu}{\tau}\right)^p e^{\frac{1}{4\tau}\Delta_C}\prod_{j=1}^{p}\frac{1}{\mu^2 - (w_j + ik_j)^2}\bigg|_{k=0} = \left(\frac{\mu}{\tau}\right)^p\left[\prod_{j=1}^{p}\frac{1}{\mu^2 - w_j^2} + \mathcal{O}\left(\frac{1}{\tau}\right)\right],$$

with $\Delta_C$ as defined in eq. (9). It follows that the effect size expectation values are, to first order in $\tau^{-1}$,

$$\mathbb{E}(x_j) = \frac{1}{2\tau}\frac{\partial\log Z}{\partial w_j} \sim \frac{1}{\tau}\frac{w_j}{\mu^2 - w_j^2},$$

which indeed converge to the minimum-energy solution $\hat{x} = 0$.

### A.6 Generalized partition functions for the expected effects

Using elementary properties of the Fourier transform, it follows that

$$\mathcal{F}\left[x_j e^{-2\tau f(x)}\right](k) = i\frac{\partial\mathcal{F}\left[e^{-2\tau f(x)}\right](k)}{\partial k_j}, \tag{10}$$

with $f$ defined in eq. (8), and hence, repeating the calculations leading up to eq. (10), we find

$$\mathbb{E}(x_j) = \frac{\int_{\mathbb{R}^p}x_j e^{-\tau H(x)}dx}{\int_{\mathbb{R}^p}e^{-\tau H(x)}dx} = \frac{Z\left[\left(C^{-1}(w-z)\right)_j\right]}{Z} \sim \left[C^{-1}(w-\hat{u}_\tau)\right]_j. \tag{11}$$

Note that eq. (10) can also be applied to the Laplacian part $e^{-2\tau g(x)}$, with $g$ defined in eq. (9). This results in

$$\mathbb{E}(x_j) = \frac{Z\left[\frac{z_j}{\tau(\mu^2 - z_j^2)}\right]}{Z} \sim \frac{\hat{u}_{\tau,j}}{\tau(\mu^2 - \hat{u}_{\tau,j}^2)}. \tag{12}$$

By the saddle point equations, eq. (13), eqs. (11) and (12) are identical. As a rule of thumb, 'tricks' such as eq. (10) to express properties of the posterior distribution as generalized partition functions lead to accurate approximations if the final result does not depend on whether the trick was applied to the Gaussian or Laplacian part of the Gibbs factor. For higher-order moments of the posterior distribution, this means that the leading term of the stationary phase approximation alone is not sufficient.

# B Proof of Theorem 1

## B.1 Saddle-point equations

Consider the function $H_\tau^*$ defined in eq. (12),

$$H_\tau^*(z) = (z - w)^T C^{-1}(z - w) - \frac{1}{\tau} \sum_{j=1}^{p} \ln(\mu^2 - z_j^2),$$

with $z$ restricted to the domain $\mathcal{D} = \{z \in \mathbb{C}^p : |\Re z_j| < \mu, \ j = 1, \ldots, p\}$. Writing $z = u + iv$, where $u$ and $v$ are the real and imaginary parts of $z$, respectively, we obtain

$$\Re H_\tau^*(z) = (u - w)^T C^{-1}(u - w) - v^T C^{-1} v - \frac{1}{2\tau} \sum_{j=1}^{p} \left\{ \ln\left[(\mu + u_j)^2 + v_j^2)\right] + \ln\left[(\mu - u_j)^2 + v_j^2)\right] \right\}$$

$$\Im H_\tau^*(z) = 2(u - w)^T C^{-1} v - \frac{1}{\tau} \sum_{j=1}^{p} \left\{ \arctan\left(\frac{v_j}{\mu + u_j}\right) + \arctan\left(\frac{v_j}{\mu - u_j}\right) \right\},$$

where $\Re c$ and $\Im c$ denote the real and imaginary parts of a complex number $c$, respectively.

By the Cauchy-Riemann equations $z = u + iv$ is a saddle point of $H_\tau^*$ if and only if it satisfies the equations

$$\frac{\partial \Re H_\tau^*}{\partial u_j} = 2[C^{-1}(u - w)]_j - \frac{1}{\tau} \left\{ \frac{\mu + u_j}{(\mu + u_j)^2 + v_j^2} - \frac{\mu - u_j}{(\mu - u_j)^2 + v_j^2} \right\} = 0$$

$$\frac{\partial \Re H_\tau^*}{\partial v_j} = -2[C^{-1} v]_j - \frac{1}{\tau} \left\{ \frac{v_j}{(\mu + u_j)^2 + v_j^2} + \frac{v_j}{(\mu - u_j)^2 + v_j^2} \right\} = 0$$

The second set of equations is solved by $v = 0$, and because $\Re H_\tau^*(u + iv) < \Re H_\tau^*(u)$ for all $u$ and $v \neq 0$, it follows that $v = 0$ is the saddle point solution. Plugging this into the first set of equations gives

$$[C^{-1}(u - w)]_j + \frac{u_j}{\tau(\mu^2 - u_j^2)} = 0, \tag{13}$$

which is equivalent to eq. (13).

## B.2 Analytic expression for the partition function

Next, consider the complex integral

$$\mathcal{I} = (-i)^p \int_{-i\infty}^{i\infty} \cdots \int_{-i\infty}^{i\infty} e^{\tau H_\tau^*(z)} Q(z) dz_1 \ldots dz_p,$$

i.e. $\mathcal{I}$ is the generalized partition function upto a constant multiplicative factor. By Cauchy's theorem we can freely deform the integration contours to a set of vertical contours running parallel to the imaginary axis and passing through the saddle point, i.e. integrate over $z = \hat{u}_\tau + ik$, where $\hat{u}_\tau$ is the saddle point solution and $k \in \mathbb{R}^p$. Changing the integration variable back from complex $z$ to real $k$, we find

$$\mathcal{I} = e^{\tau(w - \hat{u}_\tau)C^{-1}(w - \hat{u}_\tau)} \int_{\mathbb{R}^p} e^{-\tau F(k)} Q(\hat{u}_\tau + ik) dk$$

where

$$F(k) = k^T C^{-1} k - 2i k^T C^{-1}(\hat{u}_\tau - w) + \frac{1}{\tau} \sum_{j=1}^{p} \ln(\mu - \hat{u}_{\tau,j} - ik_j) + \frac{1}{\tau} \sum_{j=1}^{p} \ln(\mu + \hat{u}_{\tau,j} + ik_j).$$

We start by computing the Taylor series for $F$. First note that the $n^{\text{th}}$ derivative of $f_j^\pm(k_j) = \ln(\mu \pm \hat{u}_{\tau,j} \pm ik_j)$ evaluated at $k_j = 0$ is given by

$$(f_j^\pm)^{(n)}(0) = -\frac{(\mp i)^n (n - 1)!}{(\mu \pm \hat{u}_{\tau,j})^n}.$$

By the saddle point equations (13)

$$\frac{1}{\tau}\sum_{j=1}^{p} f_j^{+'}(0)k_j + \frac{1}{\tau}\sum_{j=1}^{p} f_j^{-'}(0)k_j = \frac{i}{\tau}\sum_{j=1}^{p}\frac{k_j}{\mu + \hat{u}_{\tau,j}} - \frac{i}{\tau}\sum_{j=1}^{p}\frac{k_j}{\mu - \hat{u}_{\tau,j}} = 2ik^T C^{-1}(\hat{u}_{\tau,j} - w).$$

Hence the linear terms cancel and we obtain

$$F(k) = \frac{1}{\tau}\sum_{j=1}^{p}\Big[\ln(\mu + \hat{u}_{\tau,j}) + \ln(\mu - \hat{u}_{\tau,j})\Big] + k^T C^{-1}k + \frac{1}{\tau}\sum_{j=1}^{p}\frac{\mu^2 + \hat{u}_{\tau,j}^2}{(\mu^2 - \hat{u}_{\tau,j}^2)^2}k_j^2$$

$$-\frac{1}{\tau}\sum_{j=1}^{p}\sum_{n\geq 3}\frac{1}{n}\Big[\frac{1}{(\mu - \hat{u}_{\tau,j})^n} + \frac{(-1)^n}{(\mu + \hat{u}_{\tau,j})^n}\Big](ik_j)^n$$

$$=\frac{1}{\tau}\sum_{j=1}^{p}\ln(\mu^2 - \hat{u}_{\tau,j}^2) + k^T(C^{-1} + D_\tau^{-1})k - \frac{1}{\tau}R_\tau(ik),$$

with $D_\tau$ the diagonal matrix defined in eq. (15) and $R_\tau$ the function defined in eq. (17). Hence

$$\mathcal{I} = e^{\tau(w-\hat{u}_\tau)C^{-1}(w-\hat{u}_\tau)}\prod_{j=1}^{p}\frac{1}{\mu^2 - \hat{u}_{\tau,j}^2}\int_{\mathbb{R}^p} e^{-\tau k^T(C^{-1}+D_\tau^{-1})k}e^{R_\tau(ik)}Q(\hat{u}_\tau + ik)dk.$$

Application of Lemma 1 results in

$$\int_{\mathbb{R}^p} e^{-\tau k^T(C^{-1}+D_\tau^{-1})k}e^{R_\tau(ik)}Q(\hat{u}_\tau + ik)dk$$

$$= \frac{(2\pi)^{\frac{p}{2}}}{(2\tau)^{\frac{p}{2}}\sqrt{\det(C^{-1}+D_\tau^{-1})}}\exp\Big\{\frac{1}{4\tau^2}\Delta_\tau\Big\}e^{R_\tau(ik)}Q(\hat{u}_\tau + ik)\Big|_{k=0}$$

$$= \Big(\frac{\pi}{\tau}\Big)^{\frac{p}{2}}\Big(\frac{\det(D_\tau)\det(C)}{\det(C+D_\tau)}\Big)^{\frac{1}{2}}\exp\Big\{\frac{1}{4\tau^2}\Delta_\tau\Big\}e^{R_\tau(ik)}Q(\hat{u}_\tau + ik)\Big|_{k=0}$$

$$= \pi^{\frac{p}{2}}\frac{\prod_j(\mu^2 - \hat{u}_{\tau,j}^2)}{\prod_j(\mu^2 + \hat{u}_{\tau,j}^2)^{\frac{1}{2}}}\Big(\frac{\det(C)}{\det(C+D_\tau)}\Big)^{\frac{1}{2}}\exp\Big\{\frac{1}{4\tau^2}\Delta_\tau\Big\}e^{R_\tau(ik)}Q(\hat{u}_\tau + ik)\Big|_{k=0},$$

where we used the equality

$$C^{-1} + D_\tau^{-1} = C^{-1}(C + D_\tau)D_\tau^{-1},$$

and $\Delta_\tau$ is the differential operator defined in eq. (16). Hence

$$Z[Q] = \frac{\mu^p}{(\pi\tau)^{\frac{p}{2}}\sqrt{\det(C)}}\mathcal{I}$$

$$= \Big(\frac{\mu}{\sqrt{\tau}}\Big)^p \frac{1}{\prod_j(\mu^2 + \hat{u}_{\tau,j}^2)^{\frac{1}{2}}}\frac{e^{\tau(w-\hat{u}_\tau)C^{-1}(w-\hat{u}_\tau)}}{\sqrt{\det(C+D_\tau)}}\exp\Big\{\frac{1}{4\tau^2}\Delta_\tau\Big\}e^{R_\tau(ik)}Q(\hat{u}_\tau + ik)\Big|_{k=0}.$$

The derivation above is formal and meant to illustrate how the various terms in the partition function approximation arise. It is rigorous if the inverse Fourier transform of $e^{R_\tau(ik)}Q(\hat{u}_\tau + ik)$ exists at least a a tempered distribution (cf. the proof of Lemma 1). This is the case if $Q$ has compact support. If this is not the case, one first has to truncate $\mathcal{I}$ to a compact region around the saddle point, and use standard estimates [5] that the contribution of the region not containing the saddle point is exponentially vanishing. Likewise, application of the operator $e^{\frac{1}{4\tau^2}\Delta_\tau}$ is defined through its series expansion, but this is to be understood as an asymptotic expansion (see below) which need not result in a convergent series. None of this is different from the standard theory for the asymptotic approximation of integrals [5].

## B.3 Asymptotic properties of the saddle point

Let $\hat{u} = \lim_{\tau\to\infty}\hat{u}_\tau$. By continuity, $\hat{u}$ is a solution to the set of equations

$$(u_j - \mu)(u_j + \mu)\big[C^{-1}(u - w)\big]_j = 0 \tag{14}$$

subject to the constraints $|u_j| \leq \mu$. Denote by $I \subseteq \{1, \ldots, p\}$ the subset of indices $j$ for which $\left[C^{-1}(\hat{u} - w)\right]_j \neq 0$. To facilitate notation, for $v \in \mathbb{R}^p$ a vector, denote by $v_I \in \mathbb{R}^{|I|}$ the sub-vector corresponding to the indices in $I$. Likewise denote by $C_I \in \mathbb{R}^{|I| \times |I|}$ the corresponding sub-matrix and by $C_I^{-1}$ the inverse of $C_I$, i.e. $C_I^{-1} = (C_I)^{-1} \neq (C^{-1})_I$. Temporarily denoting $B = C^{-1}$, we can then rewrite the equations for $\hat{u}$ as

$$\hat{u}_I = \pm \mu$$

$$\left[C^{-1}(\hat{u} - w)\right]_{I^c} = [B(\hat{u} - w)]_{I^c} = B_{I^c}(\hat{u}_{I^c} - w_{I^c}) + B_{I^c I}(\hat{u}_I - w_I) = 0,$$

or, using standard results for the inverse of a partitioned matrix [9],

$$\hat{u}_{I^c} = w_{I^c} + B_{I^c}^{-1} B_{I^c I}(w_I - \hat{u}_I) = w_{I^c} - C_{I^c I} C_I^{-1}(w_I - \hat{u}_I).$$

Finally, define $\hat{x} = C^{-1}(w - \hat{u})$, and note that

$$\hat{x}_I = [B(w - \hat{u})]_I = B_I(w_I - \hat{u}_I) + B_{II^c}(w_{I^c} - \hat{u}_{I^c}) = (B_I - B_{II^c} B_{I^c}^{-1} B_{I^c,I})(w_I - \hat{u}_I)$$

$$= C_I^{-1}(w_I - \hat{u}_I) \neq 0 \tag{15}$$

$$\hat{x}_{I^c} = 0. \tag{16}$$

As we will see below, $\hat{x} = \operatorname{argmin}_{x \in \mathbb{R}^p} H(x)$ is the maximum-likelihood lasso or elastic net solution (cf. Appendix C), and hence the set $I$ corresponds to the set of non-zero coordinates in this solution. Note that it is possible to have $\hat{u}_j = \pm \mu$ for $j \in I^c$ (i.e. $\hat{x}_j = 0$). This happens when $\mu$ is exactly at the transition value where $j$ goes from not being included to being included in the ML solution. We will denote the subsets of $I^c$ of transition and non-transition coordinates as $I_t^c$ and $I_{nt}^c$, respectively. We then have the following lemma:

**Lemma 2.** *In the limit $\tau \to \infty$, we have*

$$\tau(\mu^2 - \hat{u}_{\tau,j}^2)^2 = \begin{cases} \mathcal{O}(\tau^{-1}) & j \in I \\ \mathcal{O}\left[(\tau \hat{x}_{\tau,j}^2)^{-1}\right] & j \in I_t^c \\ \mathcal{O}(\tau) & j \in I_{nt}^c \end{cases} \tag{17}$$

*Proof.* From the saddle point equations, we have

$$\tau(\mu^2 - \hat{u}_{\tau,j}^2)^2 = \frac{1}{\tau}\left(\frac{\hat{u}_{\tau,j}}{\hat{x}_{\tau,j}}\right)^2.$$

If $j \in I$, $\hat{x}_{\tau,j} \to \hat{x}_j \neq 0$ and $\hat{u}_{\tau,j} \to \hat{u}_j = \pm\mu$, and hence $\tau(\mu^2 - \hat{u}_{\tau,j}^2)^2 = \mathcal{O}(\tau^{-1})$. If $j \in I_{nt}^c$, $\mu^2 - \hat{u}_{\tau,j}^2 \to \mu^2 - \hat{u}_j^2 > 0$, and hence $\tau(\mu^2 - \hat{u}_{\tau,j}^2)^2 = \mathcal{O}(\tau)$. If $j \in I_t^c$, $\hat{x}_{\tau,j} \to 0$ and $\hat{u}_{\tau,j} \to \hat{u}_j = \pm\mu$, and hence $\tau(\mu^2 - \hat{u}_{\tau,j}^2)^2 = \mathcal{O}\left[(\tau \hat{x}_{\tau,j}^2)^{-1}\right]$. $\qquad\square$

### B.4 Asymptotic properties of the differential operator matrix

Let

$$E_\tau = \tau D_\tau (C + D_\tau)^{-1} C = \frac{\tau}{2}\left[D_\tau(C + D_\tau)^{-1}C + C(C + D_\tau)^{-1}D_\tau\right], \tag{18}$$

where the second equality is simply to make the symmetry of $E_\tau$ explicit. We have the following result:

**Proposition 1.** *Using the block matrix notation introduced above, and assuming $I_t^c = \emptyset$, the leading term of $E_\tau$ in the limit $\tau \to \infty$ can be written as*

$$E_\tau \sim \tau \begin{pmatrix} D_{\tau,I} & \frac{1}{2}D_{\tau,I}C_I^{-1}C_{II^c} \\ \frac{1}{2}D_{\tau,I}C_I^{-1}C_{II^c} & (C^{-1})_{I^c} \end{pmatrix}, \tag{19}$$

*where $I$ is again the set of non-zero coordinates in the maximum-likelihood solution.*

*Proof.* Again using standard properties for the inverse of a partitioned matrix [9], and the fact that $D_\tau$ is a diagonal matrix, we have for any index subset $J$

$$\left[(C + D_\tau)^{-1}\right]_J = \left[C_J + D_{\tau,J} - C_{J,J^c}(C_{J^c} + D_{\tau,J^c})^{-1}C_{J^c,J}\right]^{-1} \tag{20}$$

$$\left[(C + D_\tau)^{-1}\right]_{J,J^c} = -(C_J + D_{\tau,J})^{-1}C_{J^c,J}\left[(C + D_\tau)^{-1}\right]_{J^c} \tag{21}$$

By Lemma 2, in the limit $\tau \to \infty$, $D_\tau$ vanishes on $I$ and diverges on $I^c$. Hence

$$(C_I + D_{\tau,I})^{-1} \sim C_I^{-1} \tag{22}$$

$$(C_{I^c} + D_{\tau,I^c})^{-1} \sim D_{\tau,I^c}^{-1} \tag{23}$$

Plugging these in eqs. (20) and (21), and using the fact that $C_{I,I^c} D_{\tau,I^c}^{-1} C_{I^c,I}$ is vanishingly small compared to $C_I$, yields

$$(C + D_\tau)^{-1} \sim \begin{pmatrix} C_I^{-1} & -C_I^{-1} C_{I,I^c} D_{\tau,I^c}^{-1} \\ -D_{\tau,I^c}^{-1} C_{I^c,I} C_I^{-1} & D_{\tau,I^c}^{-1} \end{pmatrix}$$

Plugging this in eq. (18), and again using that $D_{\tau,I^c}^{-1}$ is vanishingly small compared to constant matrices yields eq. (19). $\qquad\square$

From the fact that by Lemma 2, $\tau D_{\tau,I} \sim$ const, it follows immediately that, if $I_t^c = \emptyset$,

$$(E_\tau)_{ij} = \begin{cases} \mathcal{O}(\tau) & i, j \in I^c \\ \text{const} & \text{otherwise} \end{cases} \tag{24}$$

For transition coordinates, eq. (17) may diverge or not, depending on the rate of $\hat{x}_{\tau,j} \to 0$. Define

$$J = I \cup \left\{ j \in I_t^c : \lim_{\tau \to \infty} \tau^{\frac{1}{2}} \hat{x}_{\tau,j} \neq 0 \right\}. \tag{25}$$

Then $D_\tau$ diverges on $J^c$ and converges (but not necessarily vanishes) on $J$, and eqs. (22) and (23) remain valid if we use the set $J$ rather than $I$ to partition the matrix (with a small modification in eq. (22) to keep an extra possible constant term). Hence, we obtain the following modification of eq. (24):

$$(E_\tau)_{ij} = \begin{cases} \mathcal{O}(\tau) & i, j \in J^c \\ \text{const} & \text{otherwise} \end{cases} \tag{26}$$

## B.5 Asymptotic properties of the differential operator argument

Next we consider the function $R_\tau(z)$ appearing in the argument of the differential operator in eq. (14) and defined in eq. (17),

$$R_\tau(z) = \sum_{j=1}^p R_{\tau,j}(z_j)$$

$$R_{\tau,j}(z_j) = \sum_{m \geq 3} \frac{1}{m} \left[ \frac{1}{(\mu - \hat{u}_{\tau,j})^m} + \frac{(-1)^m}{(\mu + \hat{u}_{\tau,j})^m} \right] (z_j)^m.$$

We have the following result:

**Lemma 3.** $R_{\tau,j}(z_j)$ *is of the form*

$$R_{\tau,j}(z_j) = z_j^3 q_{\tau,j}(z_j)$$

*with $q_{\tau,j}$ an analytic function in a region around $z_j = 0$ and*

$$q_{\tau,j}(z_j) \leq \begin{cases} \mathcal{O}(\tau^2) & j \in J \\ \mathcal{O}(\tau) & j \in J^c \cap I_t^c \\ \text{const} & j \in I_{nt}^c \end{cases}$$

*with $J$ defined in eq. (25).*

*Proof.* The first statement follows from the fact that the series expansion of $R_{\tau,j}(z_j)$ contains only powers of $z_j$ greater than 3. The asymptotics as a function of $\tau$ for $j \in I$ and $j \in I_{nt}^c$ follow immediately from Lemma 2 and the definition of $R_{\tau,j}$ (Appendix B.2),

$$R_{\tau,j}(z_j) = -\ln\left[\mu^2 - (\hat{u}_{\tau,j} + z_j)^2\right] + \ln(\mu^2 - \hat{u}_{\tau,j}^2) - \frac{2\hat{u}_{\tau,j}}{\mu^2 - \hat{u}_{\tau,j}^2} z_j - \frac{\mu^2 + \hat{u}_{\tau,j}^2}{(\mu^2 - \hat{u}_{\tau,j}^2)^2} z_j^2.$$

For $j \in J \cap I_t^c$, we have from Lemma 2 at worst $(\mu^2 - \hat{u}_{\tau,j}^2)^{-2} = \mathcal{O}\left[(\tau \hat{x}_{\tau,j})^2\right] \leq \mathcal{O}(\tau^2)$, whereas for $j \in J^c \cap I_t^c$, we have at worst $(\tau \hat{x}_{\tau,j})^2 = \tau(\tau^{\frac{1}{2}} \hat{x}_{\tau,j})^2 \leq \mathcal{O}(\tau)$. $\qquad\square$

## B.6 Asymptotic approximation for the partition function

To prove the analytic approximation eq. (18), we will show that successive terms in the series expansion of $e^{\frac{1}{4\tau^2}\Delta_\tau}$ result in terms of decreasing power in $\tau$. The argument presented below is identical to existing proofs of the stationary phase approximation for multi-dimensional integrals [5], except that we need to track and estimate the dependence on $\tau$ in both $\Delta_\tau$ and $R_\tau$.

The series expansion of the differential operator exponential can be written as:

$$
\exp\left\{\frac{1}{4\tau^2}\Delta_\tau\right\} = \sum_{m\geq 0} \frac{1}{m!(2\tau)^{2m}} \Delta_\tau^m
$$

$$
= \sum_{m\geq 0} \frac{1}{m!(2\tau)^{2m}} \sum_{j_1,\ldots,j_{2m}=1}^{p} E_{j_1 j_2}\ldots E_{j_{2m-1}j_{2m}} \frac{\partial^{2m}}{\partial k_{j_1}\ldots\partial k_{j_{2m}}}
$$

$$
= \sum_{m\geq 0} \frac{1}{m!(2\tau)^{2m}} \sum_{\alpha:\,|\alpha|=2m} S_{\tau,\alpha} \frac{\partial^{2m}}{\partial k_1^{\alpha_1}\ldots\partial k_p^{\alpha_p}},
$$

where $E$ is the matrix defined in eq. (18) (its dependence on $\tau$ is omitted for notational simplicity), $\alpha = (\alpha_1,\ldots,\alpha_p)$ is a multi-index, $|\alpha| = \sum_j \alpha_j$, and $S_{\tau,\alpha}$ is the sum of all terms $E_{j_1 j_2}\ldots E_{j_{2m-1}j_{2m}}$ that give rise to the same multi-index $\alpha$. From eq. (26), it follows that only coordinates in $J^c$ give rise to diverging terms in $S_{\tau,\alpha}$, and only if they are coupled to other coordinates in $J^c$. Hence the total number $\sum_{j\in J^c}\alpha_j$ of $J^c$ coordinates can be divided over at most $\frac{1}{2}\sum_{j\in J^c}\alpha_j$ $E$-factors, and we have

$$
S_{\tau,\alpha} \leq \mathcal{O}\big(\tau^{\frac{1}{2}\sum_{j\in J^c}\alpha_j}\big).
$$

Turning our attention to the partial derivatives, we may assume without loss of generality that the argument function $Q$ is a finite sum of products of monomials and hence it is sufficient to prove eq. (18) with $Q$ of the form $Q(z) = \prod_{j=1}^p Q_j(z_j)$. By Cauchy's theorem and Lemma 3, we have for $\epsilon > 0$ small enough,

$$
\frac{\partial^{\alpha_j}}{\partial k_j^{\alpha_j}} e^{R_{\tau,j}(ik_j)} Q_j(ik_j)\Big|_{k_j=0} = \frac{\alpha_j!}{2\pi i}\oint_{|z|=\epsilon} \frac{1}{z^{\alpha_j+1}} e^{R_{\tau,j}(z_j)} Q_j(z_j)dz_j
$$

$$
= \frac{\alpha_j!}{2\pi i}\sum_{n\geq 0}\frac{1}{n!}\oint_{|z|=\epsilon} z_j^{3n-\alpha_j-1} q_j(z_j)^n Q_j(z_j)dz_j
$$

$$
= \frac{\alpha_j!}{2\pi i}\sum_{0\leq n<\frac{1}{3}(\alpha_j+1)}\frac{1}{n!}\oint_{|z|=\epsilon} z_j^{3n-\alpha_j-1} q_j(z_j)^n Q_j(z_j)dz
$$

$$
\leq \begin{cases} \mathcal{O}\big(\tau^{\frac{2}{3}\alpha_j}\big) & j\in J \\ \mathcal{O}\big(\tau^{\frac{1}{3}\alpha_j}\big) & j\in J^c\cap I_t^c \\ \text{const} & j\in I_{nt}^c \end{cases}
$$

The last result follows, because for $j\in J$ or $j\in J^c\cap I_t^c$, $q_j$ scales at worst as $\tau^2$ or $\tau$, respectively, and hence, since only powers of $q_j$ strictly less than $\frac{1}{3}(\alpha_j+1)$ contribute to the sum, the sum must be a polynomial in $\tau$ of degree less than $\frac{2}{3}\alpha_j$ or $\frac{1}{3}\alpha_j$, respectively ($\alpha_j$ can be written as either $3t$, $3t+1$ or $3t+2$ for some integer $t$; in all three cases, the largest integer strictly below $\frac{1}{3}(\alpha_j+1)$ equals $t$, and $t\leq\frac{1}{3}\alpha_j$).

Hence

$$
\sum_{\alpha:\,|\alpha|=2m} S_{\tau,\alpha} \frac{\partial^{2m}}{\partial k_1^{\alpha_1}\ldots\partial k_p^{\alpha_p}} e^{R_\tau(ik)}Q(ik)\Big|_{k=0} = \sum_{\alpha:\,|\alpha|=2m} S_{\tau,\alpha}\prod_j \frac{\partial^{\alpha_j}}{\partial k_j^{\alpha_j}} e^{R_{\tau,j}(ik_j)}Q_j(ik_j)\Big|_{k_j=0}
$$

$$
\leq \mathcal{O}\big(\tau^{\frac{1}{2}\sum_{j\in J^c}\alpha_j}\tau^{\frac{2}{3}\sum_{j\in J}\alpha_j+\frac{1}{3}\sum_{j\in J^c\cap I_t^c}\alpha_j}\big) = \mathcal{O}\big(\tau^{\frac{2}{3}\sum_{j\in J}\alpha_j+\frac{1}{2}\sum_{j\in I_{nt}^c}\alpha_j+\frac{5}{6}\sum_{j\in J^c\cap I_t^c}\alpha_j}\big)
$$

$$
\leq \mathcal{O}\big(\tau^{\frac{5}{6}\sum_{j=1}^p\alpha_j}\big) = \mathcal{O}\big(\tau^{\frac{5}{3}m}\big)
$$

This in turn implies that the $m^{\text{th}}$ term in the expansion,

$$\exp\left\{\frac{1}{4\tau^2}\Delta_\tau\right\}e^{R_\tau(ik)}Q(ik)\bigg|_{k=0} = \sum_{m\geq 0}\frac{1}{m!(2\tau)^{2m}}\sum_{\alpha:\,|\alpha|=2m}S_{\tau,\alpha}\prod_j\frac{\partial^{\alpha_j}}{\partial k_j^{\alpha_j}}e^{R_{\tau,j}(ik_j)}Q_j(ik_j)\bigg|_{k_j=0}$$
(27)

is bounded by a factor of $\tau^{-\frac{1}{3}m}$. Hence eq. (27) is an asymptotic expansion, with leading term

$$\exp\left\{\frac{1}{4\tau^2}\Delta_\tau\right\}e^{R_\tau(ik)}Q(ik)\bigg|_{k=0} \sim \prod_{j=1}^p Q_j(0) = Q(0).$$

$\square$

## C  Zero-temperature limit of the partition function

The connection between the analytic approximation (18) and the minimum-energy (or maximum-likelihood) solution is established by first recalling that Fenchel's convex duality theorem implies that [10]

$$\hat{x} = \underset{x\in\mathbb{R}^p}{\operatorname{argmin}}\, H(x) = \underset{x\in\mathbb{R}^p}{\operatorname{argmin}}\big[f(x) + g(x)\big] = \nabla f^*(-\hat{u}) = C^{-1}(w - \hat{u}),$$

where $f$ and $g$ are defined in eqs. (8)–(9),

$$f^*(u) = \max_{x\in\mathbb{R}^p}\big[x^T u - f(x)\big] = \frac{1}{2}(w + u)^T C^{-1}(w + u)$$

is the Legendre-Fenchel transform of $f$, and

$$\hat{u} = \underset{\{u\in\mathbb{R}^p:\,|u_j|\leq\mu,\forall j\}}{\operatorname{argmin}}\, f^*(-u) = \underset{\{u\in\mathbb{R}^p:\,|u_j|\leq\mu,\forall j\}}{\operatorname{argmin}}\,(w - u)^T C^{-1}(w - u).$$
(28)

One way of solving an optimization problem with constraints of the form $|u_j| \leq \mu$ is to approximate the hard constraints by a smooth, so-called 'logarithmic barrier function' [11], i.e. solve the unconstrained problem

$$\hat{u}_\tau = \underset{u\in\mathbb{R}^p}{\operatorname{argmin}}\left[(w - u)^T C^{-1}(w - u) - \frac{1}{\tau}\sum_{j=1}^p \ln(\mu^2 - u_j^2)\right]$$
(29)

such that in the limit $\tau \to \infty$, $\hat{u}_\tau \to \hat{u}$. Comparison with eqs. (12)–(13), shows that (29) is precisely the saddle point of the partition function, whereas the constrained optimization in eq. (28) was already encountered in eq. (14). Hence, let $I$ again denote the set of non-zero coordinates in the maximum-likelihood solution $\hat{x}$. The following result characterizes completely the partition function in the limit $\tau \to \infty$, provided there are no transition coordinates.

**Proposition 2.** *Assume that $\mu$ is not a transition value, i.e. $j \in I \Leftrightarrow \hat{x}_j \neq 0 \Leftrightarrow |\hat{u}_j| = \mu$. Let $\sigma = \operatorname{sgn}(\hat{u})$ be the vector of signs of $\hat{u}$. Then $\operatorname{sgn}(\hat{x}_I) = \sigma_I$, and*

$$Z \sim \frac{e^{\tau(w_I - \mu\sigma_I)^T C_I^{-1}(w_I - \mu\sigma_I)}}{2^{\frac{|I|}{2}}\tau^{\frac{|I|}{2}+|I^c|}\sqrt{\det(C_I)}}\prod_{j\in I^c}\frac{\mu}{\mu^2 - \hat{u}_j^2}.$$
(30)

*In particular,*

$$\lim_{\tau\to\infty}\frac{1}{\tau}\ln Z = (w_I - \mu\sigma_I)^T C_I^{-1}(w_I - \mu\sigma_I) = H(\hat{x}) = \min_{x\in\mathbb{R}^p} H(x).$$

*Proof.* First note that from the saddle point equations

$$(\mu^2 - \hat{u}_{\tau,j}^2)\hat{x}_{\tau,j} = \frac{\hat{u}_{\tau,j}}{\tau},$$

where as before $\hat{x}_\tau = C^{-1}(w - \hat{u}_\tau)$, and the fact that $|\hat{u}_{\tau,j}| < \mu$, it follows that $\operatorname{sgn}(\hat{x}_{\tau,j}) = \operatorname{sgn}(\hat{u}_{\tau,j})$ for all $j$ and all $\tau$. Let $j \in I$. Because $\hat{x}_{\tau,j} \to \hat{x}_j \neq 0$, it follows that there exists $\tau_0$ large enough

such that $\text{sgn}(\hat{x}_{\tau,j}) = \text{sgn}(\hat{x}_j)$ for all $\tau > \tau_0$. Hence also $\text{sgn}(\hat{u}_{\tau,j}) = \text{sgn}(\hat{x}_j)$ for all $\tau > \tau_0$, and since $\hat{u}_{\tau,j} \to \hat{u}_j \neq 0$, we must have $\text{sgn}(\hat{u}_j) = \text{sgn}(\hat{x}_j)$.

To prove eq. (30), we will calculate the leading term of $\det(C + D_\tau)$ in eq. (18). For this purpose, recall that for a square matrix $M$ and any index subset $I$, we have [9]

$$\det(M) = \det(M_I)\det(M_{I^c} - M_{I^c I}M_I^{-1}M_{II^c}) = \frac{\det(M_I)}{\det\big[(M^{-1})_{I^c}\big]} \tag{31}$$

Taking $M = C + D_\tau$, it follows from eqs. (20)–(23) that $\det(C_I + D_{\tau,I}) \sim \det(C_I)$, and $\det\big[(M^{-1})_{I^c}\big] \sim \det(D_{\tau,I^c}^{-1})$, and hence

$$\det(C + D_\tau) \sim \det(C_I)\det(D_{\tau,I^c}) = \tau^{|I^c|}\det(C_I)\prod_{j \in I^c}\frac{(\mu^2 - \hat{u}_{\tau,j}^2)^2}{\mu^2 + \hat{u}_{\tau,j}^2}.$$

Hence

$$\tau^{\frac{p}{2}}\prod_{j=1}^{p}\sqrt{\mu^2 + \hat{u}_{\tau,j}^2}\sqrt{\det(C + D_\tau)} \sim \tau^{\frac{p+|I^c|}{2}}\sqrt{\det(C_I)}\prod_{j \in I}\sqrt{\mu^2 + \hat{u}_{\tau,j}^2}\prod_{j \in I^c}(\mu^2 - \hat{u}_{\tau,j}^2)$$

$$\sim \tau^{\frac{p+|I^c|}{2}}2^{\frac{|I|}{2}}\mu^{|I|}\sqrt{\det(C_I)}\prod_{j \in I^c}(\mu^2 - \hat{u}_j^2),$$

where the last line follows by replacing $\hat{u}_{\tau,j}$ by its leading term $\hat{u}_j$, and using $\hat{u}_j^2 = \mu^2$ for $j \in I$. Plugging this in eq. (18) and using eqs. (15)–(16) to get the leading term of the exponential factor results in eq. (30). $\qquad\square$

The leading term in eq. (30) has a pleasing interpretation as a 'two-phase' system,

$$Z = \frac{1}{(2\pi)^{\frac{|I|}{2}}}Z_I Z_{I^c}$$

where $Z_I$ and $Z_{I^c}$ are the partition functions (normalization constants) of a multivariate Gaussian distribution and a product of independent shifted Laplace distributions, respectively:

$$Z_I = \Big(\frac{\pi}{\tau}\Big)^{\frac{|I|}{2}}\frac{e^{\tau(w_I - \mu\sigma_I)^T C_I^{-1}(w_I - \mu\sigma_I)}}{\sqrt{\det(C_I)}} = \int_{\mathbb{R}^{|I|}}e^{-\tau[x_I^T C_I x_I - 2(w_I - \mu\sigma_I)^T x_I]}dx_I$$

$$Z_{I^c} = \frac{1}{\tau^{|I^c|}}\prod_{j \in I^c}\frac{\mu}{\mu^2 - \hat{u}_j^2} = \int_{\mathbb{R}^{|I^c|}}e^{-2\tau[\mu\sum_{j \in I^c}|x_j| - \hat{u}_{I^c}^T x_{I^c}]}dx_{I^c}.$$

This suggests that in the limit $\tau \to \infty$, the non-zero maximum-likelihood coordinates are approximately normally distributed and decoupled from the zero coordinates, which each follow a shifted Laplace distribution. At finite values of $\tau$ however, this approximation is too crude, and more accurate results are obtained using the leading term of eq. (18). This is immediately clear from the fact that the partition function is a continous function of $w \in \mathbb{R}^p$, which remains true for the leading term of eq. (18), but not for eq. (30), which exhibits discontinuities whenever a coordinate enters or leaves the set $I$ as $w$ is smoothly varied.

## D   Analytic results for independent predictors

When predictors are independent, the matrix $C$ is diagonal, and the partition function can be written as a product of one-dimensional integrals

$$Z = \int_{\mathbb{R}}e^{-\tau(cx^2 - 2wx + 2\mu|x|)}dx,$$

where $c, \mu > 0$ and $w \in \mathbb{R}$. This integral can be solved by writing $Z = Z^+ + Z^-$, where

$$Z^{\pm} = \int_0^\infty e^{-\tau[cx^2 \pm 2(w \pm \mu)x]}dx = e^{\tau\frac{(w \pm \mu)^2}{c}}\int_0^\infty e^{-\tau c(x \pm \frac{w \pm \mu}{c})^2}dx = \frac{e^{\tau\frac{(w \pm \mu)^2}{c}}}{\sqrt{\tau c}}\int_{\pm\sqrt{\frac{\tau}{c}}(w \pm \mu)}^\infty e^{-y^2}dy$$

$$= \frac{1}{2}\sqrt{\frac{\pi}{\tau c}}e^{\tau\frac{(w \pm \mu)^2}{c}}\text{erfc}\Big(\pm\sqrt{\frac{\tau}{c}}(w \pm \mu)\Big) = \frac{1}{2}\sqrt{\frac{\pi}{\tau c}}\text{erfcx}\Big(\pm\sqrt{\frac{\tau}{c}}(w \pm \mu)\Big), \tag{32}$$

where $\mathrm{erfc}(x) = \frac{2}{\sqrt{\pi}}\int_x^\infty e^{-y^2}\,dy$ and $\mathrm{erfcx}(x) = e^{x^2}\,\mathrm{erfc}(x)$ are the complementary and scaled complementary error functions, respectively. Hence,

$$\log Z = \log\left[\mathrm{erfcx}\left(\sqrt{\tfrac{\tau}{c}}(\mu+w)\right) + \mathrm{erfcx}\left(\sqrt{\tfrac{\tau}{c}}(\mu-w)\right)\right] + \frac{1}{2}\left(\log\pi - \log(\tau c)\right) - \log 2,$$

and

$$
\begin{aligned}
\hat{x}_\tau = \mathbb{E}(x) &= \frac{1}{2\tau}\frac{\partial\log Z}{\partial w} \\
&= \frac{1}{c}\frac{(\mu+w)\,\mathrm{erfcx}\left(\sqrt{\tfrac{\tau}{c}}(\mu+w)\right) - (\mu-w)\,\mathrm{erfcx}\left(\sqrt{\tfrac{\tau}{c}}(\mu-w)\right)}{\mathrm{erfcx}\left(\sqrt{\tfrac{\tau}{c}}(\mu+w)\right) + \mathrm{erfcx}\left(\sqrt{\tfrac{\tau}{c}}(\mu-w)\right)} \\
&= \frac{w}{c} + \frac{\mu}{c}\frac{\mathrm{erfcx}\left(\sqrt{\tfrac{\tau}{c}}(\mu+w)\right) - \mathrm{erfcx}\left(\sqrt{\tfrac{\tau}{c}}(\mu-w)\right)}{\mathrm{erfcx}\left(\sqrt{\tfrac{\tau}{c}}(\mu+w)\right) + \mathrm{erfcx}\left(\sqrt{\tfrac{\tau}{c}}(\mu-w)\right)} \\
&= \frac{w}{c} + (1-2\alpha)\frac{\mu}{c},
\end{aligned}
$$

where

$$\alpha = \frac{1}{1 + \dfrac{\mathrm{erfcx}\left(\sqrt{\tfrac{\tau}{c}}(\mu-w)\right)}{\mathrm{erfcx}\left(\sqrt{\tfrac{\tau}{c}}(\mu+w)\right)}}.$$

# E   Numerical recipes

## E.1   Solving the saddle point equations

To calculate the partition function and posterior distribution at any value of $\tau$, we need to solve the set of equations in eq. (13). To avoid having to calculate the inverse matrix $C^{-1}$, we make a change of variables $x = C^{-1}(w-u)$, or $u = w - Cx$, such that eq. (13) becomes

$$x_j\left[w_j - (Cx)_j - \mu\right]\left[w_j - (Cx)_j + \mu\right] + \frac{1}{\tau}\left[w_j - (Cx)_j\right] = 0. \tag{33}$$

We will use a coordinate descent algorithm where one coordinate of $x$ is updated at a time, using the current estimates $\hat{x}$ for the other coordinates. Defining

$$a_j = w_j - \sum_{k\neq j} C_{kj}\hat{x}_k,$$

we can write eq. (33) as

$$C_{jj}^2 x_j^3 - 2a_j C_{jj} x_j^2 + \left(a_j^2 - \mu^2 - \frac{C_{jj}}{\tau}\right)x_j + \frac{a_j}{\tau} = 0$$

The roots of this 3rd order polynomial are easily obtained numerically, and by construction there will be a unique root for which $u_j = w_j - (Cx)_j = a_j - C_{jj}x_j$ is located in the interval $(-\mu, \mu)$. This root will be the new estimate $\hat{x}_j$. Given a new $\hat{x}_j^{(\mathrm{new})}$, we can update the vector $a$ as

$$a_k^{(\mathrm{new})} = \begin{cases} a_j^{(\mathrm{old})} & k = j \\ a_k^{(\mathrm{old})} - C_{kj}\left(\hat{x}_j^{(\mathrm{new})} - \hat{x}_j^{(\mathrm{old})}\right) & k \neq j \end{cases}$$

and proceed to update the next coordinate.

After all coordinates of $\hat{x}$ have converged, we obtain $\hat{u}_\tau$ by performing the matrix-vector operation

$$\hat{u}_\tau = w - C\hat{x},$$

or, if we only need the expectation values,

$$\mathbb{E}_\tau(x) = \hat{x}.$$

For $\tau = \infty$, the solution to eq. (33) is given by the maximum-likelihood effect size vector (cf. Appendix C), for which ultra-fast algorithms exploiting the sparsity of the solution are available [12]. Hence we use this vector as the initial vector for the coordinate descent algorithm for $\tau < \infty$ and expect fast convergence if $\tau$ is large. Solutions for multiple values of $\tau$ can be obtained along a descending path of $\tau$-values, each time taking the previous solution as the initial vector for finding the next solution.

## E.2 High-dimensional determinants in the partition function

Calculating the stationary phase approximation to the partition function involves the computation of the $p$-dimensional determinant $\det(C + D_\tau)$ [cf. eq. (18)], which can become computationally expensive in high-dimensional settings. However, when $C$ is of the form $C = \frac{A^T K^{-1} A}{2n} + \lambda \mathbb{1}$ [cf. eq. (6)] with $A \in \mathbb{R}^{n \times p}$, $K \in \mathbb{R}^{n \times n}$ invertible, and $p > n$, these determinants can be written as $n$-dimensional determinants, using the matrix determinant lemma:

$$\det(C + D_\tau) = \det\Big(\frac{A^T K^{-1} A}{2n} + D'_\tau\Big) = \frac{\det(D'_\tau)}{\det(K)} \det\Big(K + \frac{A(D'_\tau)^{-1} A^T}{2n}\Big), \qquad (34)$$

where $D'_\tau = D_\tau + \lambda \mathbb{1}$ is a diagonal matrix whose determinant and inverse are trivial to obtain.

To avoid numerical overflow or underflow, all calculations are performed using logarithms of partition functions. For $n$ large, a numerically stable computation of eq. (34) uses the equality $\log \det B = \operatorname{tr} \log B = \sum_{i=1}^n \log \epsilon_i$, where $B = K + \frac{1}{2n} A(D'_\tau)^{-1} A^T$ and $\epsilon_i$ are the eigenvalues of $B$.

## E.3 Marginal posterior distributions

Calculating the marginal posterior distributions $p(x_j)$ [eq. 20] requires applying the analytic approximation eq. (14) using a different $\hat{u}_\tau$ for every different value of $x_j$. To make this process more efficient, two simple properties are exploited:

1. For $x_j = \hat{x}_{\tau,j}$, the saddle point for the $(p-1)$-dimensional partition function $Z(C_{I_j}, w_{I_j} - x_j C_{j,I_j}, \mu)$ is given by the original saddle point vector $\hat{x}_{\tau,k}$, $k \neq j$. This follows easily from the saddle point equations.

2. If $x_j$ changes by a small amount, the new saddle point also changes by a small amount. Hence, taking the current saddle point vector for $x_j$ as the starting vector for solving the set of saddle point equations for the next value $x_j + \delta$ results in rapid convergence (often in a single loop over all coordinates).

Hence we always start by computing $p(x_j = \hat{x}_{\tau,j})$ and then compute $p(x_j)$ separately for a series of ascending values $x_j > \hat{x}_{\tau,j}$ and a series of descending values $x_j < \hat{x}_{\tau,j}$

## E.4 Sampling from the one-dimensional distribution

Consider again the case of one predictor, with posterior distribution

$$p(x) = \frac{e^{-\tau(cx^2 - 2wx + 2\mu|x|)}}{Z}. \qquad (35)$$

To sample from this distribution, note that

$$p(x) = (1 - \alpha)\, p(x \mid x < 0) + \alpha\, p(x \mid x \geq 0),$$

where

$$p(x \mid x \in \mathbb{R}^\pm) = \frac{e^{-\tau(cx^2 - 2(w \mp \mu)x)}}{Z^\mp}, \qquad (36)$$

$Z^\pm$ were defined in eq. (32), and

$$\alpha = P(x \geq 0) = \int_0^\infty p(x)dx = \frac{1}{Z}\int_0^\infty e^{-\tau[cx^2 - 2(w-\mu)x]}dx = \frac{Z^-}{Z} = \frac{1}{1 + \frac{\operatorname{erfcx}\left(\sqrt{\frac{\tau}{c}}(\mu - w)\right)}{\operatorname{erfcx}\left(\sqrt{\frac{\tau}{c}}(\mu + w)\right)}}.$$

Eq. (36) defines two truncated normal distributions with means $(w \mp \mu)/c$ and standard deviation $1/\sqrt{2\tau c}$, for which sampling functions are available. Hence, to sample from the distribution (35), we first sample a Bernoulli random variable with probability $\alpha$, and then sample from the appropriate truncated normal distribution.

## E.5 Gibbs sampler

To sample from the Gibbs distribution in the general case, we use the 'basic Gibbs sampler' of [13]. Let $\hat{x}$ be the current vector of sampled regression coefficients. Then a new coefficient $x_j$ is sampled from the conditional distribution

$$p\big(x_j \mid \{\hat{x}_k, k \neq j\}\big) = \frac{e^{-\tau[C_{jj}x_j^2 - 2a_j x_j + 2\mu|x_j|]}}{Z_j}, \tag{37}$$

where $a_j = w_j - \sum_{k \neq j} C_{kj}\hat{x}_k$ and $Z_j$ is a normalization constant. This distribution is of the same form as eq. (35) and hence can be sampled from in the same way. Notice that, as in section E.1, after sampling a new $\hat{x}_j$, we can update the vector $a$ as

$$a_k^{(\text{new})} = \begin{cases} a_j^{(\text{old})} & k = j \\ a_k^{(\text{old})} - C_{kj}\big(\hat{x}_j^{(\text{new})} - \hat{x}_j^{(\text{old})}\big) & k \neq j \end{cases}.$$

## E.6 Maximum a-posteriori estimation of the inverse temperature

This paper is concerned with the problem of obtaining the posterior regression coefficient distribution for the Bayesian lasso and elastic net when values for the hyperparameters $(\lambda, \mu, \tau)$ are given. There is abundant literature on how to select values for $\lambda$ and $\mu$ for maximum-likelihood estimation, mainly through cross validation or by predetermining a specific level of sparsity (i.e. number of non-zero predictors). Hence we assume an appropriate choice for $\lambda$ and $\mu$ has been made, and propose to then set $\tau$ equal to a first-order approximation of its maximum a posteriori (MAP) value, i.e. finding the value which maximizes the log-likelihood of observing data $y \in \mathbb{R}^n$ and $A \in \mathbb{R}^p$, similar to what was suggested by [13]. To do so we must include the normalization constants in the prior distributions (2)–(3):

$$p(y \mid A, x, \tau) = \left(\frac{\tau}{2\pi n}\right)^{\frac{n}{2}} e^{-\frac{\tau}{2n}\|y - Ax\|^2} = \left(\frac{\tau}{2\pi n}\right)^{\frac{n}{2}} e^{-\frac{\tau}{2n}\|y\|^2} e^{-\frac{\tau}{2n}[x^T A^T A x - 2(A^T y)^T x]}$$

$$p(x \mid \lambda, \mu, \tau) = \frac{e^{-\tau(\lambda\|x\|^2 + 2\mu\sum_j |x_j|)}}{Z_0}$$

where for $\lambda > 0$,

$$Z_0 = \int_{\mathbb{R}^p} dx \, e^{-\tau(\lambda\|x\|^2 + 2\mu\sum_j |x_j|)} = \left(\int_{\mathbb{R}} dx \, e^{-\tau(\lambda x^2 + 2\mu|x|)}\right)^p = \left(2\int_0^\infty dx \, e^{-\tau(\lambda x^2 + 2\mu x)}\right)^p$$

$$= \left(\frac{2e^{\frac{\mu^2\tau}{\lambda}}}{\sqrt{\lambda\tau}} \int_{\sqrt{\frac{\mu^2\tau}{\lambda}}}^\infty e^{-t^2} dt\right)^p = \left(\sqrt{\frac{\pi}{\lambda\tau}} e^{\frac{\mu^2\tau}{\lambda}} \text{erfc}\left(\sqrt{\frac{\mu^2\tau}{\lambda}}\right)\right)^p \sim \left(\frac{1}{\mu\tau}\right)^p, \tag{38}$$

and the last relation follows from the first-order term in the asymptotic expansion of the complementary error function for large values of its argument,

$$\text{erfc}(x) \sim \frac{e^{-x^2}}{x\sqrt{\pi}}.$$

For pure lasso regression ($\lambda = 0$), this relation is exact:

$$Z_0 = \left(\frac{1}{\mu\tau}\right)^p.$$

Hence, the log-likelihood of observing data $y \in \mathbb{R}^n$ and $A \in \mathbb{R}^p$ given values for $\lambda, \mu, \tau$ is

$$\mathcal{L} = \log \int_{\mathbb{R}^p} dx \, p(y \mid A, x, \tau) p(x \mid \lambda, \mu, \tau)$$

$$= \frac{n}{2}\log\tau - \frac{\|y\|^2}{2n}\tau - \log Z_0 + \log \int_{\mathbb{R}^p} dx \, e^{-\tau H(x)} + \text{const},$$

where 'const' are constant terms not involving the hyperparameters. Taking the first order approximation

$$\log Z = \log \int_{\mathbb{R}^p} dx \, e^{-\tau H(x)} \sim -\tau H_{\min} = -\tau H(\hat{x}),$$

where $\hat{x}$ are the maximum-likelihood regression coefficients, we obtain

$$\mathcal{L} \sim \left(p + \frac{n}{2}\right) \log \tau - \left[\frac{\|y\|^2}{2n} + H(\hat{x})\right] \tau + p \log \mu$$

$$= \left(p + \frac{n}{2}\right) \log \tau - \left[\frac{1}{2n}\|y - A\hat{x}\|^2 + \lambda\|\hat{x}\|^2 + 2\mu\|\hat{x}\|_1\right] \tau + p \log \mu$$

which is maximized at

$$\tau = \frac{p + \frac{n}{2}}{\frac{1}{2n}\|y - A\hat{x}\|^2 + \lambda\|\hat{x}\|^2 + 2\mu\|\hat{x}\|_1}.$$

Note that a similar approach to determine the MAP value for $\lambda$ would require keeping an additional second order term in eq. (38), and that for $p > n$ it is not possible to simultaneously determine MAP values for all three hyperparameters, because it leads to a set of equations that are solved by the combination $\lambda = \mu = 0$ and $\tau = \infty$.

# F   Experimental details

## F.1   Hardware and software

All numerical experiments were performed on a standard Macbook Pro with 2.8 GHz processor amd 16 GB RAM running macOS version 10.13.6 and Matlab version R2018a.   Maximum-likelihood elastic net models were fitted using Glmnet for Matlab (`https://web.stanford.edu/~hastie/glmnet_matlab/`).   Matlab software to solve the saddle point equations, compute the partition function and marginal posterior distributions, and run a Gibbs sampler, is available at `https://github.com/tmichoel/bayonet/`.   Bayesian horseshoe and an alternative Bayesian lasso Gibbs sampler were run using the BayesReg toolbox for Matlab [14], available at `https://uk.mathworks.com/matlabcentral/fileexchange/60823-bayesian-penalized-regression-with-continuous-shrinkage-prior-densities`.

## F.2   Diabetes and leukemia data

The diabetes data were obtained from `https://web.stanford.edu/~hastie/CASI_files/DATA/diabetes.html`. The leukemia data were obtained from `https://web.stanford.edu/~hastie/CASI_files/DATA/leukemia.html`. Data were standardized according to eq. (1), and no further processing was performed. For the results in Figure 2, $\lambda$ was set to $0.1$, $\mu$ was selected as the smallest value with a maximum-likelihood solution with 5 (diabetes data) or 10 (leukemia data) non-zero predictors, and $\tau$ was set to its maximum a-posteriori value given $\lambda$ and $\mu$, yielding $\tau = 682.3$ (diabetes data) and $9.9439 \cdot 10^3$ (leukemia data).

## F.3   Cancer Cell Line Encyclopedia data

Normalized expression data for 18,926 genes in 917 cancer cell lines were obtained from the Gene Expression Omnibus accession number GSE36139 using the Series Matrix File `GSE36139-GPL15308_series_matrix.txt`. Drug sensitivity data for 24 compounds in 504 cell lines were obtained from the supplementary material of [15] (tab 11 from supplementary file `nature11003-s3.xls`); 474 cell lines were common between gene expression and drug response data and used for our analyses. Of the available drug response data, only the activity area ('actarea') variable was used; 7 compounds had more than 40 zero activity area values (meaning inactive compounds) in the 474 cell lines and were discarded. For the remaining 17 compounds, the following procedure was used to compare the stationary phase approximation for the Bayesian elastic net to BayReg's lasso and horseshoe regression methods and maximum-likelihood elastic net and ridge regression:

1. For each response variable (drug), possible hyper-parameter values were set to $\lambda = 0.1$ (fixed); $\mu_n = \mu_{\max} \times r^{\frac{N+1-n}{N}}$, where $N = 10$, $n = 1, \ldots, 10$, $r = 0.01$ and $\mu_{\max} = \max_{j=1,\ldots,p} |w_j|$, with $w$ as defined in eq. (6)–(7) and $p = 18,926$; $\tau_m = 10^{0.25(m+M-1)}$, where $M = 12$, $m = 1, 2, \ldots, 13$.

2. For each training data set, and for each drug, the following procedure was performed:

(a) The 1,000 genes most strongly correlated with the response were selected as candidate predictors.

(b) Response and predictor data were standardized.

(c) Ten-fold cross-validation was carried out, by randomly dividing the 427 training samples in 10 sets of 42 samples for testing; running Bayesian and maximum-likelihood elastic net on the remaining 385 samples and evaluating predictive performance (correlation of predicted and true drug sensitivities) on the test set. The values $\hat{\mu}_{ML}, \hat{\mu}_{BAY}$ nd $\hat{\tau}_{BAY}$ with best median performance were selected.

(d) Maximum-likelihood coefficients for ridge regression ($\lambda = 0.1, \mu = 0$) and elastic net regression ($\lambda = 0.1, \mu = \hat{\mu}_{ML}$) were calculated on the training set (427 samples).

(e) Bayesian posterior expectation values using the stationary phase approximation for elastic net ($\lambda = 0.1, \mu = \hat{\mu}_{ML}, \tau = \hat{\tau}_{BAY}$) were calculated on the training set (427 samples).

(f) Bayesian posterior expectation values for lasso and horseshoe regression using Gibbs sampling (using BayReg with default parameter settings) were calculated on the training set (427 samples).

(g) Drug responses were predicted on the original data scale in the 47 held-out validation samples using all sets of regression coefficients, and the Pearson correlation with the true drug response was calculated.

3. For each drug, the median correlation value over the 10 predictions was taken, resulting in the values shown in Figure 3a.

The top 1,000 most correlated genes were pre-filtered in each training data set, partly because in trial runs this resulted in better predictive performance than pre-selecting 5,000 or 10,000 genes, and partly to speed up calculations.

Figure 3d shows the regression coefficients for the training fold whose performance was closest to the median.

For Figure 3c, Bayesian posterior expectation values using the stationary phase approximation for elastic net and maximum-likelihood regression coefficients were calculated on each training set (427 samples) over a denser grid of 20 values for $\mu$ (same formula as above with $N - 20$) and the same 13 values for $\tau$, and evaluated on the 47 validation samples; the median correlation value of predicted and true drug sensitivities over the ten folds is shown. Figure 3b shows the dependence on $\mu$ of the maximum-likelihood performance, and performance of the best $\tau$ at every $\mu$ for the Bayesian method.

# G   Supplementary figures

Figure S1: Illustration of the stationary phase approximation procedure for $p = 1$. **(a)** Contour plot of the complex function $(z - w)^2$. If $\mu = \mu_2$, the integration contour can be deformed from the imaginary axis to a steepest descent contour parallel to the imaginary axis and passing through the saddle point $z_0 = w$, whereas if $\mu = \mu_1$, this cannot be done without passing through the pole at $z = \mu$. **(b,c)** Contour plots of the complex function $(z - w)^2 - \frac{1}{\tau} \ln(\mu^2 - z^2)$ for $|w| < \mu$ and $|w| \geq \mu$, respectively. In both cases the function has a unique saddle point $u_\tau$ with $|u_\tau| < \mu$ and a steepest descent contour that is locally parallel to the imaginary axis.

Figure S2: Same as Figure 3b and c, for drugs 2–9 from Figure 3a.

Figure S3: Same as Figure 3b and c, for drugs 10–17 from Figure 3a.