[Reviews · NeurIPS 2018]

Reviewer 1



I think this is a very well written paper. The approach presented aim at using an approximation in the case of Bayesian lasso and elastic net, so to speed up the standard Gibbs sampling. These two cases are difficult to be managed with other approximations, e.g. the Laplace approximation, because the likelihood is not double differentiable. The authors propose to use a stationary phase approximation such that the intractable normalising constant is expressed in terms of Fourier transform. This approximation reduce the computational costs of a standard Gibbs sampling, in particular in the presence of a large number of predictors. The approach is well described within the available literature and the results, including advantages and disadvantages, are clearly presented. The approach is interesting, even if a set of saddle point equations have to be solved, and a balance exists between goodness of the approximation and computational speed. I have few comments. There is a focus on comparing the results with the maximum-likelihood coefficients: I do not understand completely this choice, since a Bayesian approach may be different in the context of variable selection. I may have missed this part, but the setting of Figure 3 is not totally clear: how the true drug sensitivity is computed? In the last part of Section 3, an insight on how to perform prediction is presented. However I do not agree that the focus of prediction is often an expected value. Is there a way to produce an evaluation of the uncertainty within this approach?

Reviewer 2



Summary An approximation to the posterior distribution from a Bayesian lasso or Bayesian elastic net prior is developed. The method uses a saddle-point approximation to the partition function. This is developed by writing the posterior distribution in terms of tau = n / sigma^2 and uses an approximation for large tau. The results are illustrated on three data sets: diabetes (n=442, p=10), leukaemia (n=72, p=3571) and Cancer Cell Line Encyclopedia (n=474, p=1000). These demonstrate some of the performance characteristics of the approximation. Strengths The paper is clearly written and shows a new, useful method for approximation posterior distributions in sparse regression problems. This is an original approach and there is a wealth of background information in the appendix. There is strong evidence that the method works well on a range of problems. Posterior inference with sparsity priors in linear regression is an important problem and a good approximation would be very useful. As the author notes Gibbs sampling can be slow (although there have been advances in this area for large p, small n problems, see e.g. Johndrow et al, 2018). This paper provides a big step in providing a fast optimisation based approach. Weaknesses My reading of the paper is that the method is limited to linear regression with prior distributions whose log density is convex. It would interesting to know whether there is any scope to extend to logistic regression models (or other GLMs) or to other sparse priors (such as the horseshoe). I found it difficult to interpret the results in Figure 1. What is a small error in this context? The results suggest that tau needs to be at least 100000 for an extremely small error but, presumably, the method works well with smaller values of tau (for example, the diabetes data uses tau=682.3). A discussion would be useful. For example, a small n (say approx n=100) and low signal-to-noise ratio could easily lead to tau in 100's. Would the method work well in these situations? Page 7 discusses prediction using the posterior predictive mean. How easily could the posterior predictive density be calculated? Also could other summaries such as posterior variance be calculated using this approach? References J. E. Johndrow, P. Orenstein and A. Bhattacharya (2018): Scalable MCMC for Bayes Shrinkage Priors, arXiv:1705.00841

Reviewer 3



In eq.(18) of Theorem 1, the author derives the formula by which the generalized partition function can be analytically approximated. However, in this theorem, the assumption C>0 has no meaning in machine learning and statistics, because, in the Bayesian LASSO and and similar methods, C contains many zero and almost zero eigenvalues. This is the central point in the research of LASSO. Even in such a case, several statistical methods for calculating the partition function have already be constructed. Neither the conventional Laplace approximation, Gaussian integral, nor saddle point approximation can be employed, however, the new algebraic and geometrical method gives us the answers. Sumio Watanabe, A widely applicable Bayesian information criterion, Journal of Machine Learning Research, Vol.14, (Mar), pp.867-897, 2013. Mathias Drton, Martyn Plummer, A Bayesian information criterion for singular models, Royal Statistical Society Series B, doi.org/10.1111/rssb.12187, 2017